# Connectivity concepts in neuronal network modeling

**Johanna Senk**[1]*, **Birgit Kriener**[2], **Mikael Djurfeldt**[3], **Nicole Voges**[4], **Han-Jia Jiang**[1,5], **Lisa Schüttler**[6], **Gabriele Gramelsberger**[6], **Markus Diesmann**[1,7,8], **Hans E. Plesser**[1,9], **Sacha J. van Albada**[1,5]

**1** Institute of Neuroscience and Medicine (INM-6) and Institute for Advanced Simulation (IAS-6) and JARA-Institut Brain Structure-Function Relationships (INM-10), Jülich Research Centre, Jülich, Germany, **2** Institute of Basic Medical Sciences, University of Oslo, Oslo, Norway, **3** PDC Center for High-Performance Computing, KTH Royal Institute of Technology, Stockholm, Sweden, **4** INT UMR 7289, Aix-Marseille University, Marseille, France, **5** Institute of Zoology, University of Cologne, Cologne, Germany, **6** Chair of Theory of Science and Technology, Human Technology Center, RWTH Aachen University, Aachen, Germany, **7** Department of Psychiatry, Psychotherapy and Psychosomatics, School of Medicine, RWTH Aachen University, Aachen, Germany, **8** Department of Physics, Faculty 1, RWTH Aachen University, Aachen, Germany, **9** Faculty of Science and Technology, Norwegian University of Life Sciences, Ås, Norway

* j.senk@fz-juelich.de

**Data Availability Statement:** The code to reproduce all figures of this manuscript is publicly available at Zenodo: https://doi.org/10.5281/zenodo.5550351.

**Funding:** This project has received funding from the European Union's Horizon 2020 Framework

## Abstract

Sustainable research on computational models of neuronal networks requires published models to be understandable, reproducible, and extendable. Missing details or ambiguities about mathematical concepts and assumptions, algorithmic implementations, or parameterizations hinder progress. Such flaws are unfortunately frequent and one reason is a lack of readily applicable standards and tools for model description. Our work aims to advance complete and concise descriptions of network connectivity but also to guide the implementation of connection routines in simulation software and neuromorphic hardware systems. We first review models made available by the computational neuroscience community in the repositories ModelDB and Open Source Brain, and investigate the corresponding connectivity structures and their descriptions in both manuscript and code. The review comprises the connectivity of networks with diverse levels of neuroanatomical detail and exposes how connectivity is abstracted in existing description languages and simulator interfaces. We find that a substantial proportion of the published descriptions of connectivity is ambiguous. Based on this review, we derive a set of connectivity concepts for deterministically and probabilistically connected networks and also address networks embedded in metric space. Beside these mathematical and textual guidelines, we propose a unified graphical notation for network diagrams to facilitate an intuitive understanding of network properties. Examples of representative network models demonstrate the practical use of the ideas. We hope that the proposed standardizations will contribute to unambiguous descriptions and reproducible implementations of neuronal network connectivity in computational neuroscience.

Programme for Research and Innovation under Specific Grant Agreement 720270 (HBP SGA1) [to MDj, MDi, HEP], 785907 (HBP SGA2) [to JS, MDj, MDi, HEP, SvA], 945539 (HBP SGA3) [to JS, MDj, HJJ, MDi, HEP, SvA], and 754304 (DEEP-EST) [to HEP]; the Deutsche Forschungsgemeinschaft (DFG, German Research Foundation) - 368482240/GRK2416: "RTG 2416 Multi-senses Multi-scales" [to MDi]; the Priority Program (SPP 2041 "Computational Connectomics") of the Deutsche Forschungsgemeinschaft [to SvA]; the Helmholtz Association Initiative and Networking Fund under project number SO-092 (Advanced Computing Architectures, ACA) [to JS, MDi]; the Excellence Initiative of the German federal and state governments (neuroIC001): "ERS: disziplinärer Paketantrag NeuroIC: NeuroModelingTalk (NMT) Approaching the complexity barrier in neuroscientific modeling" [to JS, LS, GG, MDi]; and the Helmholtz Metadata Collaboration (HMC), an incubator platform of the Helmholtz Association within the framework of the Information and Data Science strategic initiative, under the funding ZT-I-PF-3-026 [to JS]. Open access publication funded by the Deutsche Forschungsgemeinschaft (DFG, German Research Foundation) - 491111487. The funders had no role in study design, data collection and analysis, decision to publish, or preparation of the manuscript.

**Competing interests:** The authors have declared that no competing interests exist.

## Author summary

Neuronal network models are simplified and abstract representations of biological brains that allow researchers to study the influence of network connectivity on the dynamics in a controlled environment. Which neurons in a network are connected is determined by connectivity rules and even small differences between rules may lead to qualitatively different network dynamics. These rules either specify explicit pairs of source and target neurons or describe the connectivity on a statistical level abstracted from neuroanatomical data. We review articles describing models together with their implementations published in community repositories and find that incomplete and imprecise descriptions of connectivity are common. Our study proposes guidelines for the unambiguous description of network connectivity by formalizing the connectivity concepts already in use in the computational neuroscience community. Further we propose a graphical notation for network diagrams unifying existing diagram styles. These guidelines serve as a reference for future descriptions of connectivity and facilitate the reproduction of insights obtained with a model as well as its further use.

## Introduction

The connectivity structure of a neuronal network model is sometimes described with a statement such as "$N_s$ source neurons and $N_t$ target neurons are connected randomly with connection probability $p$". One interpretation of this statement is an algorithm that considers each possible pair of source and target neurons exactly once and connects each such pair with probability $p$. Other interpretations of the same statement may allow multiple connections between the same pair of neurons, apply the connection probability non-uniformly on different neuron pairs, or include further assumptions on the distribution of in- and outgoing connections per neuron. These choices do not just affect the network structure, but can have substantial consequences for the network dynamics. To illustrate this point, we simulate two balanced recurrent networks of randomly connected excitatory and inhibitory spiking neurons based on the model of Brunel [1] (see Section "Materials and methods" for model details). Fig 1A shows the dynamics of the original model described in [1], where the number of incoming connections per neuron (*in-degree*) is fixed to $K_{in}$. In contrast, Fig 1B shows the dynamics of a network in which the number of outgoing connections per neuron (*out-degree*) is fixed to $K_{out}$. The total number of connections is the same in both networks and, by implication, an interpretation of the network's connection probability, too. The network-averaged spike rate has a similar pattern across time in both instantiations. However, while the rates of individual neurons are alike for the network with fixed in-degree, they are broadly distributed for the network with fixed out-degree. These small and comparatively simple example network simulations already demonstrate that ambiguities in network descriptions can result in networks with statistically different activities.

For more complex networks with spatial embedding, hierarchical organization, or higher specificity of connections, the task of fully specifying the connectivity becomes correspondingly more daunting. As researchers are building more complete models of the brain, simultaneously explaining a larger set of its properties, the number of such complex models is steadily increasing. This increase is accelerated by the rise of large-scale scientific projects which carefully assemble rich connectivity graphs. For example, the Allen Institute for Brain Science has published a model of mouse primary visual cortex with a layered structure, multiple cell types, and specific connectivity based on spatial distance and orientation preference [2]. The Blue

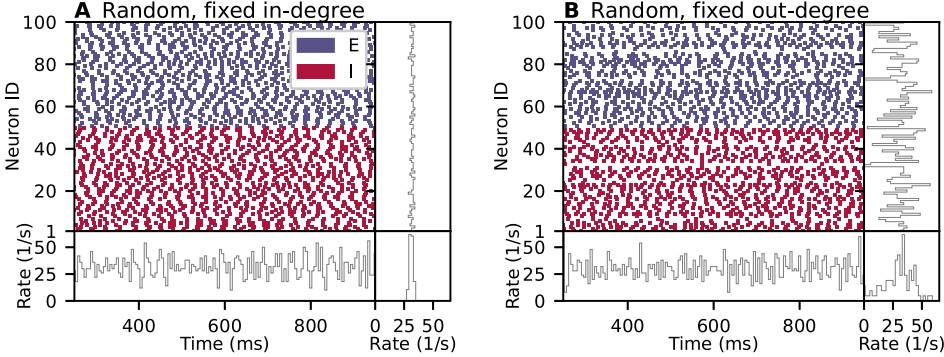

**Fig 1. Spiking neuron network simulations of a balanced random network with (A) fixed in-degree and (B) fixed out-degree.** Top left: Raster plots show spike times of 50 out of 10, 000 excitatory (E) and 50 out of 2, 500 inhibitory (I) neurons. Bottom left: Time-resolved spike rate from spike-count histogram across time with temporal bin width of 5 ms. Top right: Per-neuron spike rate from spike-count histogram for individual neurons. Bottom right: Normalized distribution of per-neuron spike rates with bin width of 2/s. Model details are given in Section "Materials and methods".

Brain microcircuit reproduces a small patch of rat somatosensory cortex featuring cell-type-specific connectivity based on paired recordings and morphological neuron reconstructions [3, 4]. The multi-area spiking network model of macaque visual cortex by Schmidt et al. [5] is a multi-scale network with specific connectivity between 32 cortical areas, each composed of interconnected excitatory and inhibitory neurons in four cortical layers. The network structure of these models is typically specified by a combination of explicit connectivity based on neuro-anatomical data and connection patterns captured by probabilistic or deterministic rules. Regardless of how connectivity is specified, reproducible research requires unambiguous network descriptions and corresponding algorithmic implementations.

Mathematically defined models of neuronal networks are to be distinguished from their concrete implementation and execution in the form of simulations. Any given model has uniquely defined dynamics apart from potential stochasticity; model variants can obviously exist, but each variant is a model in its own right. The dynamics of all but the simplest models can only be fully explored using simulations, i.e., requiring the instantiation and execution of the model in the form of a computer program. Any abstract model can be implemented in multiple ways. A central challenge in computational neuroscience, as well as other fields relying on simulations, is to define abstract models so precisely that the researcher only needs to decide how to implement the model, but not what to implement. Our focus in this work is on facilitating such precise model descriptions, particularly with regard to network connectivity.

First, we review some terminology. Model neuronal networks generally consist of nodes, which represent individual neurons or neural populations; the latter is common in models describing activity in terms of average firing rates. In a concrete simulation code, network nodes are typically first created with a dedicated command. Network nodes are connected by edges. Connections are typically directed, i.e., signals flow from a source node to a target node. When nodes represent individual neurons, edges represent one or a small number of individual synapses, and when nodes represent groups of neurons, edges represent an average over many synapses. We use the term *connection* to mean a single, atomic edge between network nodes. Neuronal network simulation software usually provides a command allowing one to create such an edge between any two network nodes.

In many models, nodes are grouped into populations of homologous neurons. Populations can be nested hierarchically, e.g., one may consider an entire brain area as a population, the neurons within a specific layer of that area, or all neurons of a given cell type within the layer. Also edges in a network can be grouped, reflecting anatomical structure (nerve bundles), purpose (inhibitory recurrent connections), or developmental processes. We call such groups of edges *projections*. They play an important role in specifying and instantiating models: We can specify network connectivity by providing, for each projection between any pair of populations, a *connection rule* which defines how to create atomic edges (connections) between individual nodes. A projection is thus defined by a triplet of source population, target population and connection rule and represents a collection of atomic connections.

Neuronal network simulation software commonly provides support for connecting populations based on connection rules, which may be deterministic or probabilistic. A key challenge in the field of computational neuroscience, which we address here, is to precisely define connections rules and their properties, so that model descriptions obtain a unique interpretation and can be matched precisely to the implementations of these rules provided by simulation software.

A command to instantiate a single model neuron of a given type and a command to create an atomic edge between any pair of neurons is all that is required to construct a neuronal network model of arbitrary complexity in a computer—the model implementer just has to arrange for the right combination of calls through loops and other control structures. However, this approach has two significant shortcomings. First, most information about the structure of the network is lost. As the network is described on the lowest possible level, terms describing higher-order organizational principles of brain structures such as cell populations, layers, areas, and projections between them do not occur; they are implicitly contained in the algorithms. This limits the readability of the model specification and thereby the ability to verify and reuse the code. It also precludes systematic visualization or exploration of the network with computational tools. Second, a simulation engine reading the code will have little opportunity to parallelize network construction. Network specifications at higher conceptual levels, on the other hand, leave a simulation engine the freedom to use efficient parallelization, for example when connecting two populations of neurons in an all-to-all fashion. With the progress of neuroscience towards larger and more structured networks, the degree of parallelization becomes relevant. In typical simulations, network creation can easily become the dominant component of the total simulation time and may hinder a research project because of the forbidding compute resources it would require [6, 7]. High-level connectivity descriptions can help by exposing organizational principles for the simulator to exploit and giving the neuroscientist access to the expert knowledge encoded in the simulator design and the reliability of code used in many peer-reviewed studies. To be useful to computational neuroscientists, connectivity concepts for neuronal network models should encompass connectivity patterns occurring in real brains. On the one hand, small brains of simple organisms such as *C. elegans* exhibit highly specific connection patterns [8], which tend to require explicit connectivity matrices for their specification. The brains of more complex organisms such as mammals, on the other hand, have a multi-scale organization that can be captured at different levels of abstraction. Their brains are divided into multiple regions, each of which may contain different neuron types forming populations with statistically similar connectivity patterns. Some regions, such as the cerebellar cortex, have highly stereotyped, repetitive connectivity motifs [9]. Elsewhere, for instance in the cerebral cortex, the neuron-level connectivity appears more random [10, 11]. Nevertheless, the cerebral cortex exhibits a number of organizational principles, including a laminar and columnar architecture. On a larger spatial scale, the cortex is subdivided into different functional areas. Each of these areas is often in itself a complex,

hierarchically structured network of substructures. These structural elements may be connected to each other, resulting in connectivity across various spatial scales.

At a basic level of organization, pairs of neurons are connected with a probability that depends on both the source and the target area and population. For instance, neurons within the same cortical layer are generally more likely to be connected to each other than neurons located in different layers [2, 12–14]. Neurons can synapse on themselves [15] and can establish more than one synapse on any given target neuron [16]. Connection probability decays with distance both at the level of areas [17, 18] and at the level of individual neurons. Within local cortical circuits, the length constant for this decay is on the order of 150–300 $\mu$m [19, 20]. Typical assumptions for the local connectivity are a Gaussian or exponential decay of the connection probability between pairs of neurons with increasing distance between their cell bodies [21, 22]. Both within and between cortical areas, excitatory neurons form so-called patchy connections consisting of spatially clustered synapses [23–26]. Within areas, this patchiness becomes apparent at the medium distance range of millimeters. Another important organizing principle is that neurons exhibit like-to-like connectivity. For instance, neurons with more similar receptive fields are more likely to be connected [27–30]. In addition, having common neighbors increases the chance for a pair of neurons or areas to be connected, also known as the homophily principle [31]. Such homophily results in the presence of connection motifs of three or more neurons beyond what would be expected based on pairwise connection probabilities alone [32]. At higher levels of organization, the cerebral cortex has a hierarchically modular structure [33]. Sometimes cortex is also described as having small-world properties [34, 35]. In our treatment of connectivity concepts, we focus on the most fundamental properties of network circuitry but also touch upon such more complex organizational aspects.

With on the order of $10^4$ incoming connections to each of the $10^{10}$ neurons of human cortex [36, 37], the estimated total number of connections in the full cortical network is $10^{14}$. Only the study of natural-density, full-scale networks gives reliable information about features such as the structure of pairwise correlations in the brain's neuronal activity [38]. Downscaled networks obtained by reducing neuron and synapse numbers may only preserve some characteristics of the network dynamics, for example the firing rates, if parameters are adjusted for compensation. In the present study, we describe connectivity concepts based on the principles of neuroanatomical organization, abstracted in a way that allows for mathematical formalization and algorithmic implementations in simulators. The concepts target both the connectivity of small proof-of-concept network models with only hundreds or thousands of interconnected neurons and large-scale networks approaching the full size and connection density of biological brains. In this endeavor, we take into account the current practice in the field by considering published models and corresponding open-source code. These resources provide insight into the connectivity types relevant to computational neuroscientists and the way in which these are described and implemented. Our aim is to derive a unified vocabulary, along with mathematical and graphical notations, for describing connectivity in a concise and non-ambiguous way. Besides supporting the reproducibility, sharing, and reuse of neuronal network models, this effort facilitates efficient implementations of high-level connectivity routines in dedicated simulation software and hardware. Here, we use the term "high-level" to refer to the abstraction of network connectivity patterns to mathematical functions of few parameters. It is possible for a network model to be partly described by such high-level connectivity, whereas other aspects of the connectivity are specified in detail. The combined connectivity of such a model can then have many parameters. Abstractions of network organization encode our understanding of the structure of the system, enable more systematic analyses, in some cases direct comparisons with analytical theory, and greater comparability between models.

The concepts we discuss specify the connectivity between network nodes that are most often individual neurons but may equally well be neural populations or brain regions. While the nodes can also be multi-compartment neurons, we are not concerned with detailed connectivity below the neuronal level such as to specific dendritic compartments. In the case of plastic networks, we only consider the initial state, and do not describe the evolution of the connectivity.

We first review published models to identify which network structures are used by the community and how they are described. Next we turn to description languages and simulators to review how connectivity is abstracted in simulation interfaces. Based on this dual review, the following section proposes connectivity concepts for deterministic and probabilistic networks, and also addresses networks embedded in metric space. In addition to these mathematical and textual descriptions of the concepts, we propose a graphical notation for illustrating network structures. Our results conclude with a few examples of how the connectivity of neuronal network models is concisely and unambiguously described and displayed using our notation. Finally we discuss our results in the context of the evolution of the field.

Preliminary work has been published in abstract form [39, 40].

## Results

### Networks used in the computational neuroscience community

We review network models for which both a manuscript and an implementation have been published. Models in computational neuroscience are often made available via one of a few common repositories. We select the most prominent repositories relevant to the present study, and in the following characterize the models fitting our scope contained in them.

The models entering this study are in the online repositories ModelDB [41, 42] and Open Source Brain (OSB) [43]. Both repositories have been maintained for years (or even decades in the case of ModelDB) and they support the curation, further development, visualization, and simulation of a large variety of models in different ways. ModelDB stores its models using the infrastructure of SenseLab (http://senselab.med.yale.edu). Implementations on ModelDB generally aim to serve as static reference for a published article (although some entries link to version-controlled repositories) and no restrictions on programming languages or simulators are made. In contrast, all models indexed by OSB (https://www.opensourcebrain.org) are stored in public version-controlled repositories such as GitHub (https://github.com) to foster ongoing and collaborative model development. Models in OSB are standardized in the sense that they are made available in the model description languages NeuroML [44, 45] or PyNN [46], besides potentially further versions.

As this study focuses on network connectivity, we review network models of point neurons, simple multicompartment neurons (without considering connectivity specific to compartments), and neural mass models, but exclude neural field models as well as more detailed models. Therefore, we narrow the broad collection of models in ModelDB down to the MicrocircuitDB section *Connectionist Networks* (https://senselab.med.yale.edu/MicroCircuitDB/ModelList.cshtml?id=83551).

Spiking, binary, and rate neurons are all accepted as network nodes. Plastic networks in which the connection strengths (e.g., spike-timing dependent plasticity [47]) or even the connectivity itself (structural plasticity [48]) evolve over time are not *a priori* excluded. However, for plastic networks we only consider structure independent of dynamics, i.e., only the result of the initial network construction. If an article describes multiple different network models, we concentrate on the one most relevant for this study. Only connections between neuronal populations are taken into account; connections with external stimulating and recording

devices are ignored. For some of the indexed publications, the full (network) model is not actually available in the repository, and we exclude such incomplete models from this study.

All selected network models are characterized based on five main and several sub-categories and the results are summarized in Figs 2–6. For the main categories, we formulate the following guiding questions:

**Metadata** (Fig 2) When, where, and by whom were article and code published?

**Description** (Fig 3) How does the article describe the connectivity and is the description complete?

**Implementation** (Fig 4) How is the connectivity technically implemented?

**Network** (Fig 5) How are network nodes and edges characterized?

**Concepts** (Fig 6) Which connectivity concepts are realized?

Our model review comprises a total of 42 selected models with about 80% of the code found in ModelDB and about 20% in OSB (Fig 2A). The corresponding articles are listed in Section "Reviewed network models" in "Materials and methods". They have appeared in a number of peer-reviewed journals and were published between 1996 and 2020; approximately 70% of the models were published since 2013 (Fig 2B and 2C). Scientists increasingly appreciate the value of reproducible research, which leads to more published code and in particular

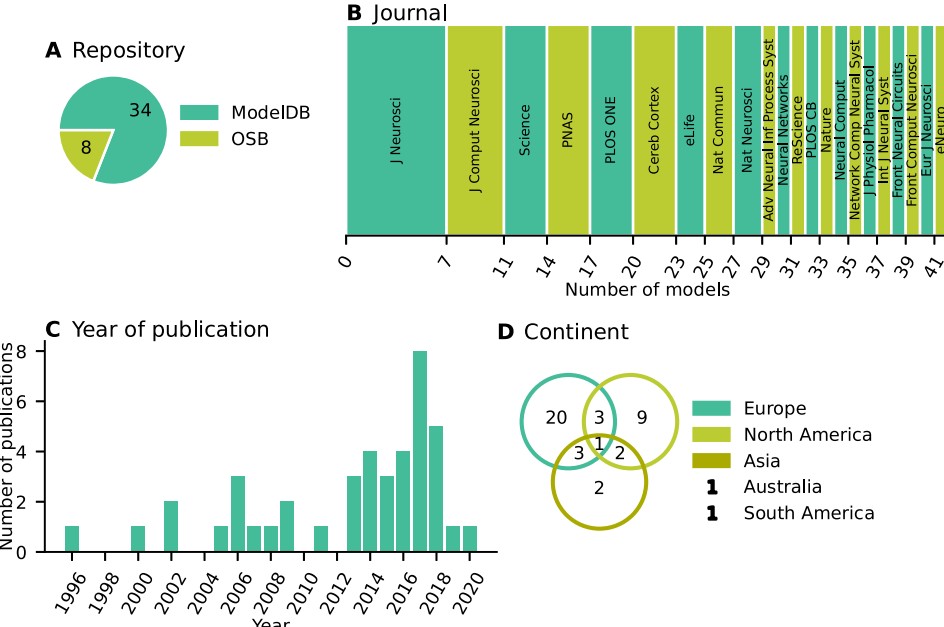

**Fig 2. Metadata: When, where, and by whom were article and code published?** (A) Pie chart of repositories storing model code. "ModelDB": section Microcircuit DB *Connectionist Networks* of ModelDB. "OSB": Open Source Brain. (B) Abbreviated journal name in stacked, horizontal bar plot. (C) Year of publication in bar plot. (D) Location of all authors' labs based on affiliations as Venn diagram. Intersections indicate collaborations between labs situated on different continents. Not included in the diagram are two publications of which all authors are affiliated with labs only in Australia and South America, respectively.

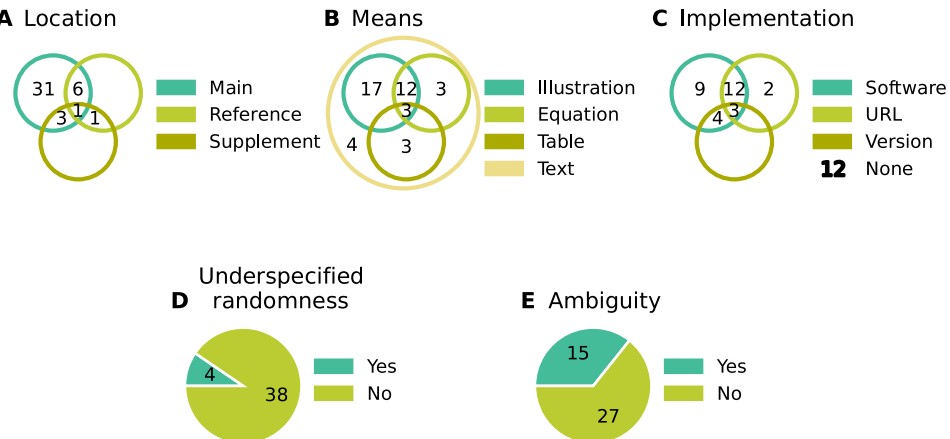

**Fig 3. Description: How does the article describe the connectivity and is the description complete?** (A) Location of connectivity description. "Main": in main manuscript; "Reference": reference to other publication; "Supplement": in separate file belonging to the same publication. (B) Means used to describe connectivity. Descriptions of the parameterization of connections are only counted if they are crucial for understanding whether connections exist. (C) Reference to model implementation in manuscript. "Software": name of software given; "URL": explicit hyperlink or DOI referencing published code; "Version": software version given; "None": implementation not mentioned (number of occurrences given in legend). Intersections in panels A–C mean that the connectivity is described in different locations, a combination of different means is used, and different references to the model implementation are given, respectively. (D) Whether connectivity is just specified as "random" or a connection probability is given without defining the connection rule. (E) Whether description is insufficient or inconclusive for implementing the network model.

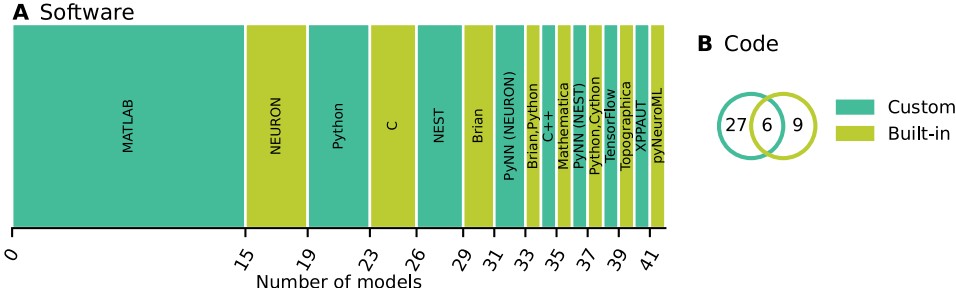

**Fig 4. Implementation: How is the connectivity technically implemented?** (A) Name of software framework (dedicated simulator or general-purpose software). (B) Implementation of connections. "Custom": hard-coded; "Built-in": routine from dedicated simulator. The intersection means that a part of the network connectivity is explicitly coded in a general-purpose language and another part uses built-in simulator functionality.

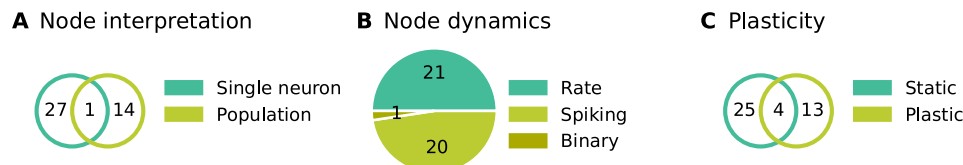

**Fig 5. Network: How are network nodes and edges characterized?** (A) Interpretation of network nodes. "Single neuron": connections exist between single neuronal units; "Population": connections are established between nodes that represent multiple neurons. (B) Dynamics of the nodes. "Rate": continuous signal; "Spiking": spiking mechanism; "Binary": on-off mechanism. (C) Plasticity. "Static": identity of connections and weight values fixed; "Plastic": potential changes of connections and weights during simulation. The intersections in panels A and C refer to models which have both properties in different parts of the networks.

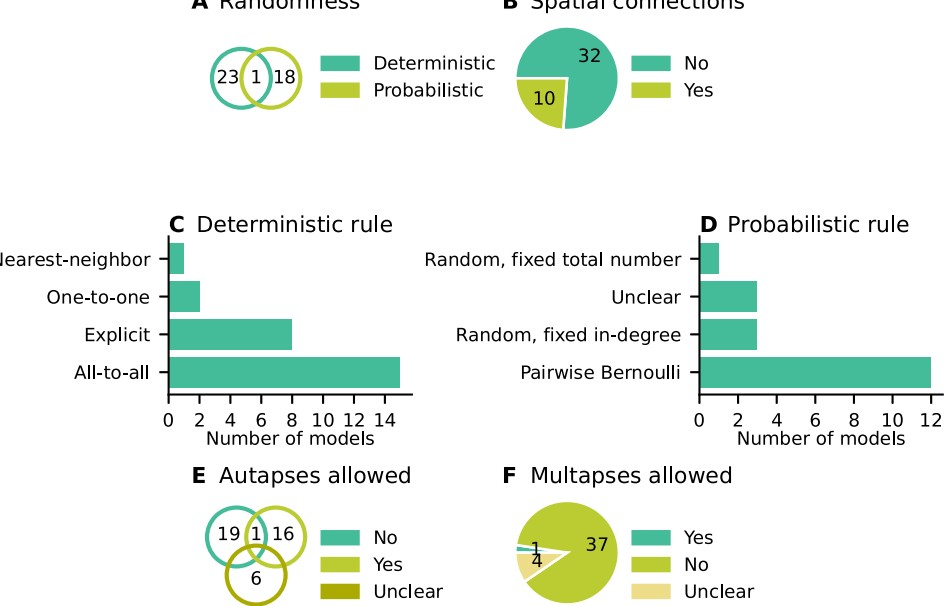

**Fig 6. Concepts: Which connectivity concepts are realized?** (A) Whether connections in the model are probabilistic or deterministic. (B) Whether at least some part of the model contains distance-dependent connections. (C) Name of deterministic connectivity rule specifying the connectivity in at least a part of the model network (compare Fig 7A and 7B). (D) Name of probabilistic connectivity rule specifying the connectivity in at least a part of the model network (compare Fig 7C–7F). One network model can use multiple deterministic and probabilistic rules or may use none of the given rules; therefore the numbers of models in panels C and D do not add up to the total number of studies. (E) Whether self-connections are allowed (illustrated in Fig 7G). The intersections in panels A, B, and E refer to models which have different properties in different parts of the networks. (F) Whether multiple connections from a source node to a target node are allowed (illustrated in Fig 7H).

more use of dedicated repositories [42, 43, 49–52]. Journal policies also play a role, as some journals explicitly encourage or even enforce the publication of code. For instance, with seven occurrences the Journal of Neuroscience (https://www.jneurosci.org) is overrepresented in our list of journals (Fig 2B) and a possible explanation is that journal's recommendation to deposit code of new computational models in suitable repositories such as ModelDB. The analysis of the authors' affiliations shows that the models under consideration were developed mostly through collaborations spanning a small number of different labs, mainly from Europe and North America (Fig 2D).

Each article studied describes the model connectivity to some degree (Fig 3A). But about a quarter of the models are described partially outside the article proper, namely in referenced publications or supplementary material. One reason for an incomplete description in the main article might be space restrictions by the journal. Another reason is that some models build on previously published ones and therefore the authors decide to state only the differences to the original model. Without exception all articles use text to describe the connectivity; mostly the text is combined with other means such as illustrations, equations, and tables (Fig 3B). These other means may be only supportive, as is often the case with illustrations, or necessary to convey the complete connectivity. Although not encountered in the studies considered here, another means of description may be code or pseudo-code. The majority of articles contain some information about the model implementation. By model implementation we mean the executable model description defined either via an interface to a dedicated neuronal network simulator or via a general-purpose programming language. More than a third of the

publications provide a direct link or other reference to the published code (Fig 3C). Since usage and default values of a software may change in the course of development, giving the software name but not the software version with which the model code needs to be run can be insufficient. More than a quarter of the articles considered do not mention a model implementation at all. We find that one reason for this observation is that the authors published the code after the article; another reason is that the published implementation occasionally does not originate from the authors of the article.

Next, we ask whether randomness in the connectivity is underspecified, meaning that either the word "random" is used without further specification, or a connection probability is given without definition (Fig 3D). This underspecification is identified in almost 10% of the articles. We find more than a third of the descriptions ambiguous (Fig 3E) due to missing details or imprecise formulations. We consider a connectivity description to be unambiguous if 1) in the case of deterministic connectivity, it enables reconstructing the identity of all connections in the model; or 2) in the case of probabilistic connectivity, it enables determining either the connectivity distribution, or the exact algorithm by which the connections were drawn. Here, we focus on the identity of the connections, including their directionality, and not on their parameterization (e.g., weights and delays).

Turning from the connectivity description in articles to the model implementations, we find that a wide variety of software is used for implementing the connectivity (Fig 4A). This software is either a general-purpose programming language such as MATLAB, Python, or C/C++, or a dedicated simulator for neuronal networks such as NEURON, NEST, or Brian. The prevalence of code for the commercial closed-source interpreter MATLAB (more than a third) may be explained by the fact that it is widely used in many research labs for analyzing experimental data and therefore has a tradition in neuroscience. Almost 80% of the model codes use custom, *ad hoc* implementations for defining the connectivity instead of, or in addition to, high-level functions provided by simulators (Fig 4B). Also precomputed or loaded adjacency matrices fall into the category "custom".

In the following, we characterize the model networks according to their node and edge properties since these affect the interpretation of connectivity. If the connectivity is defined between single neurons, a connection may represent a single synapse or several individual synapses. However, if the connectivity is defined between nodes that represent populations of neurons, a connection is rather understood as an average over multiple synapses, i.e., an effective connection. This type of connectivity exists in one third of the studied models (Fig 5A). About half of the networks use as nodes rate neurons with continuous dynamics (Fig 5B); rate dynamics often coincide with the interpretation of nodes as neural populations. The other half use spiking neurons, i.e., neuron models which integrate their inputs and fire action potentials if a threshold is crossed. We encounter only one study using binary neurons that switch between On and Off states. About 40% of the models included have plastic connections at least in some part of the network (Fig 5C). Since changes in the connection structure or the weights occur during the course of the simulation, we only take the initial connectivity into account when identifying connectivity concepts.

Fig 6 combines the connectivity description in the articles with the available model implementations to bring forward which connectivity concepts are actually realized in the studies. Those properties which remain underspecified are marked with "Unclear". The number of occurrences of "Unclear" does not add up to the number of connectivity descriptions identified as ambiguous Fig 3E. Reasons are that 1) in some cases the ambiguity in the description concerns an aspect not covered by the categories of Fig 6 (e.g., the number of connections is fixed, but the number is not given), and 2) sometimes, ambiguity in the description is solved by clear code. More than half of the models use only deterministic connection rules, and in the

other half the connections are created using probabilistic rules (Fig 6A); one model combines both deterministic and probabilistic rules. Fig 7 illustrates connectivity patterns reflecting the most common rules: the deterministic rules "one-to-one" and "all-to-all", and the probabilistic rules "random, fixed in-degree", "random, fixed total number", and "pairwise Bernoulli". Among the deterministic rules, "all-to-all" dominates in the studies considered here (Fig 6C). About a quarter of the networks included here use spatial connections in at least some part of the model network, meaning that the nodes are embedded in a metric space and the connections depend on the relative positions of source and target neurons (Fig 6B). Connections that could be described as "one-to-all" or "all-to-one" are here summarized in the more general "all-to-all". In particular the plastic network models included tend to use "all-to-all" connectivity for the initial network state and then let the weights evolve. In the networks with population-model nodes, pairs of source and target nodes were connected one at a time. Looking at this as a high-level connectivity can only be done by considering the network as a whole; it then corresponds to the rule with an explicit adjacency list, and we thus classify these cases as "explicit". "Nearest-neighbor" connectivity could be seen as a special case of "one-to-one", but we mention it here explicitly. By far the most common probabilistic rule is "pairwise Bernoulli": for each pair of nodes at most one connection is created with a given probability (Fig 6D). The second most common rule is "random, fixed in-degree". Examples for most of the remaining patterns depicted in Fig 7 are also observed, albeit in smaller numbers. Note that matched forward and reverse connections between pairs of neurons occur with deterministic rules such as "all-to-all" by construction but can also occur by chance with probabilistic rules. In one case, we encounter gap junctions which are symmetric by definition of the synapse model. *Autapses* or self-connections [53] are not allowed or do not occur by construction in about half of the networks (Fig 6E). *Multapses*, which are multiple connections between the same pair of nodes [54, 55], are allowed only in a single study (Fig 6F). We define a multapse

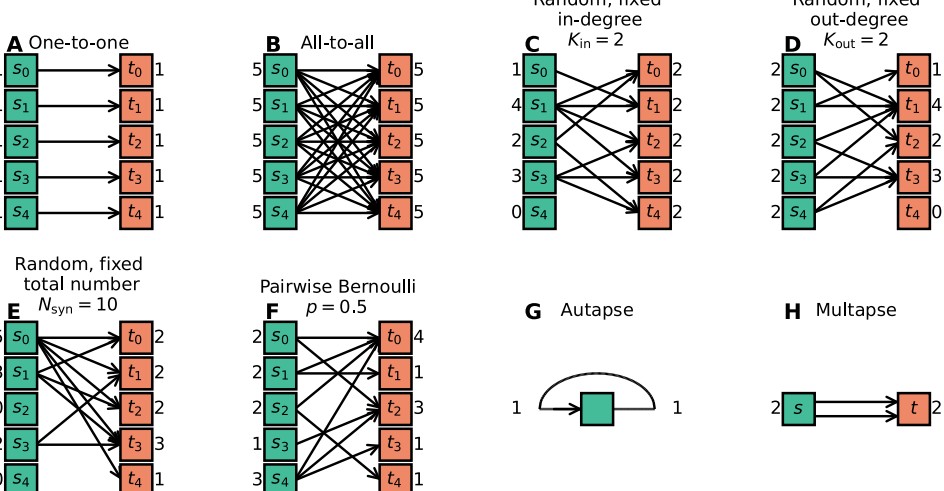

**Fig 7. Connectivity patterns reflecting the most common rules.** The ordered set of sources $\mathcal{S}$ is depicted by the green squares on the left. They are connected to the ordered set of targets $\mathcal{T}$, depicted by the orange squares on the right. The respective in- and out-degrees are given next to the nodes. (A) One-to-one. (B) All-to-all. (C) Random, fixed in-degree with $K_{in}$ connections per target node. (D) Random, fixed out-degree with $K_{out}$ connections per source node. (E) Random, fixed total number of connections $N_{syn}$. (F) Pairwise Bernoulli with connection probability $p$. (G) Autapse (self-connection). (H) Multapse (multi-connection).

as a set of connections sharing the same source node and target node and therefore also the directionality. The individual connections of a multapse can, however, use different parameters such as weights and delays. In judging the presence of multapses, a few subtleties are involved. First, cases where modelers capture the effects of multiple biological synapses using single, strong model synapses are not identified. Second, even if multiple connections between a given source and target node are explicitly generated, their effects may be lumped in the low-level code of a simulator when the model dynamics is linear [56, Section 5.3]. Autapses and multapses are rarely discussed explicitly, but their presence can be inferred from other specifications: The "pairwise Bernoulli" rule, for instance, considers each pair of nodes for connection only once; multapses are thus excluded.

## Description languages and simulators

A neuronal network simulator typically provides an idiosyncratic model description language or supports a pre-existing one, for example a cross-simulator description language like PyNN [46], NeuroML [44], NineML [57], CSA [58], or SONATA [59]. A less common case is where the simulator consists of a library with an API called by a general-purpose language such as is the case for SPLIT [60] and, to some extent, GeNN [61]. We here consider model description languages either tied to a particular simulator or supported by multiple simulators.

The ways in which network connectivity is described in such languages broadly fall into three main categories: *procedural* descriptions, *declarative* descriptions at a population-to-population level, and more general declarative descriptions using algebra. Some languages support more than one of these paradigms.

**Procedural descriptions.** Most simulators provide a primitive for connecting a source neuron to a target neuron:

```
Connect(source, target)
```

Typically, `source` and `target` above refer to indices in some neuron enumeration scheme. For example, both NEST [62–64], NEURON [65, 66] and Arbor [67, 68] have the concept of a *global identifier* or *GID* which associates a unique integer with each neuron. Many simulation environments offer a generic programming language where it is possible to write algorithms based on `Connect` to describe network connectivity. For example, the all-to-all connectivity pattern shown in Fig 7B, where each source neuron is connected to every target neuron, could be achieved by the procedural description:

```
for target in targetIndices:
    for source in sourceIndices:
        Connect(source, target)
```

A common pattern in such algorithms is to loop over all possible connections, as above, and call connect only if some condition is fulfilled.

```
for target in targetIndices:
    for source in sourceIndices:
        if condition:
            Connect(source, target)
```

If `condition` above is `random() < p`, where `random()` returns a uniformly

distributed random number $r$, $0 \leq r < 1$, we obtain the pairwise Bernoulli pattern with probability $p$ as shown in Fig 7E.

Procedural descriptions are the most general form of network specification: Any kind of connectivity pattern can be created by combining suitable `Connect` calls. Procedural descriptions of connectivity at the neuron-to-neuron level are for instance supported by the simulators NEST [62–64], NEURON [65, 66], Arbor [67, 68], Brian [69], Moose [70], and Nengo [71], as well as the description language PyNN [46].

Our example for the procedural approach already exposes two shortcomings. First, the explicit loop over all possible combinations is generic, but it is also costly if the condition is only fulfilled in a small fraction of the cases. For particular distributions an expert of statistics may know a more efficient method to create connections according to the desired distribution. Taking the example of a Bernoulli trial for each source-target pair, this knowledge can be encoded in a simulator function `pairwise_bernoulli()`. In this way also non-experts can create Bernoulli networks efficiently. Second, the explicit loops describe a serial process down to the primitive `Connect()` between two nodes. This gives simulators little chance to efficiently parallelize network construction.

**Declarative population-level descriptions.** A declarative description of connectivity describes the connectivity at a conceptual level: It focuses on the connectivity pattern we want to obtain instead of the individual steps required to create it. Typically, the declarative description names a connectivity *rule* which is then used in setting up connectivity between two neuronal populations or from a population to itself. A common example is:

```
Connect ( sourcePopulation , targetPopulation ,
        {'rule': 'all_to_all'})
```

Declarative descriptions operating on populations are expressive, since they specify connectivity in terms of generic rules. Simulator software can be optimized for each rule, especially through parallelization. Rule-based specification of connectivity helps the reader of the model description to understand the structure of the network and also allows visualization and exploration of network structure using suitable tools. Usually the user is limited to a set of predefined rules, although some simulation software allows users to add new rules.

Declarative population-level descriptions are for instance supported by the simulators NEST, Moose, and the description languages PyNN (connectors) and NineML. Commonly provided connectivity rules are: one-to-one, all-to-all, and variants of probabilistic rules. The "transforms" in Nengo can also be regarded as declarative descriptions of connectivity. NeuroML supports lists of individual connections. Its associated language NetworkML [44] provides declarative descriptions akin to those of PyNN and NineML, while its associated lower-level declarative language LEMS [45] supports the definition of types of connectivity based on partially procedural constructs ("structure element types") such as `ForEach` and `If`, giving greater flexibility but departing to some extent from the spirit of declarative description.

**Algebraic descriptions.** Using algebraic expressions to describe connectivity rivals procedural descriptions in the sense that they are generic and expressive. Such descriptions are also declarative, with the advantage of facilitating optimization and parallelization.

The Connection Set Algebra (CSA) [58] is an algebra over sets of connections. It provides a number of elementary connection sets as well as operators on them. In CSA, connectivity can be concisely described using expressions of set algebra. Implementations demonstrate that the

corresponding network connectivity can be instantiated in an efficient and parallelizable manner [72].

In CSA, a connection is represented by a pair of indices $(i, j)$ which refer to the entities being connected, usually neurons. A source population of neurons $\mathcal{S}$ can be enumerated by a bijection to a set of indices selected among the non-negative integers

$$E_{\mathcal{S}} : \mathcal{S} \to \mathbb{N}_S \subseteq \mathbb{N}_0, \tag{1}$$

and a target population $\mathcal{T}$ can be similarly enumerated. A connection pattern can then be described as a set of pairs of indices. For example, if $\mathcal{S} = \{a, b\}$, $\mathcal{T} = \{c, d\}$ and both sets of neurons are enumerated from 0 to 1, a connection pattern consisting of a connection from $a$ to $c$ and a connection from $b$ to $d$ would in CSA be represented by $\{(0, 0), (1, 1)\}$.

However, in CSA it turns out to be fruitful to work with infinite sets of indices. E.g., the *elementary* (predefined) connection set $\delta = \{(0, 0), (1, 1), \ldots\}$ can be used to describe one-to-one connectivity in general, regardless of source and target population size. We can work with CSA *operators* on infinite connection sets and extract the actual, finite, connection pattern at the end. Given the finite index sets above, $\mathbb{N}_S = \mathbb{N}_T = \{0, 1\}$, we can extract the finite one-to-one connection pattern between $\mathcal{S}$ and $\mathcal{T}$ through the expression $\delta \cap (\mathbb{N}_S \times \mathbb{N}_T)$ where $\cap$ is the set intersection operator and $\times$ is the Cartesian product.

Another example of an elementary connection set is the set of all connections

$$\Omega = \bigcup_{i,j \in \mathbb{N}_0} \{(i, j)\} = \{(0, 0), (0, 1), \ldots, (1, 0), (1, 1), \ldots\}. \tag{2}$$

For the case of connections within a population (i.e., $\mathcal{S} = \mathcal{T}$) it is now possible to create the set of all-to-all connectivity without self-connections:

$$\Omega - \delta \tag{3}$$

where $-$ is the set difference operator.

Random pairwise Bernoulli connectivity can be described by the elementary parameterized connection set $\rho(p)$, which contains each connection in $\Omega$ with probability $p$. The *random selection operator* $\rho_{\mathbf{N}}(n)$ picks $n$ connections without replacement from the set it operates on, while the operators $\rho_{\mathbf{0}}(k)$ and $\rho_{\mathbf{1}}(k)$ randomly pick connections to fulfill a given out-degree or in-degree $k$, respectively.

Multapses are treated by allowing *multisets*, i.e., multiple instances of the same connection are allowed in the set. The CSA expression for random connectivity with a total number of $n$ connections, without multapses, is:

$$\rho_{\mathbf{N}}(n)(\mathbb{N}_S \times \mathbb{N}_T) \tag{4}$$

where the Cartesian product of the source and target index sets, $\mathbb{N}_S \times \mathbb{N}_T$, constitutes the possible neuron pairs to choose from.

By instead selecting from a multiset, we can allow up to $m$ multapses:

$$\rho_{\mathbf{N}}(n) \biguplus_{i=1}^{m} (\mathbb{N}_S \times \mathbb{N}_T) \tag{5}$$

where $\biguplus$ is the multiset union operator.

The operator

$$\mathbf{M} = \biguplus_{i=1}^{\infty} \tag{6}$$

replaces each connection in a set $C$ with an infinity of the same connection such that, e.g., $\rho_0(k)\mathbf{M}C$ means picking connections in $C$ to fulfill fan-out $k$, but now effectively *with* replacement. Without going into the details, multisets can also be employed to set limits on the number of multapses.

**Population-level connectivity rules of languages and simulators.** Most neural network description languages and simulators provide several descriptors or routines that can be used to specify standard connectivity patterns in a concise and reproducible manner. We here give an overview over the corresponding connection rules offered by a number of prominent model description languages and simulators. This brief review supplements the literature review to identify a set of common rules to be more formally described in the next section.

We have studied connectivity rules of the following model specification languages and simulators:

**NEST** is a simulator which provides a range of pre-defined connection rules supporting network models with and without spatial structure. To create rule-based connections, the user provides source and target population, and the connection rule with applicable parameters and specifications of the synapses to be created, including rules for the parameterization of synapses. The information here pertains to NEST version 3.0.

In addition to the built-in connectivity rules, NEST provides an interface to external libraries, for example CSA, to specify connectivity.

**PyNN** is a simulator-independent language. It provides a class of high-level connectivity primitives called `Connector`. The connector class represents the connectivity rule to use when setting up a `Projection` between two populations. The information here pertains to PyNN version 0.9.6.

**NetPyNE** is a Python package to facilitate the development, simulation, parallelization, analysis, and optimization of biological neuronal networks using the NEURON simulator. It provides connectivity rules for explicitly defined populations as well as subsets of neurons matching certain criteria. A connectivity rule is specified using a `connParams` dictionary containing both parameters defining the set of presynaptic and postsynaptic cells and parameters determining the connectivity pattern. The information here pertains to NetPyNE version 1.0.

**NineML** is an XML-based cross-simulator model specification language.

**Brian** is a simulator which has a unique way of setting up connectivity. Connections between a source and target group of neurons are specified using an expression that combines procedural and algebraic aspects, passed to the connect method of the synapse object S:

```
S.connect(j='EXPR for VAR in RANGE if COND')
```

Here, `EXPR` is an integer-valued expression specifying the targets for a given neuron $i$. This expression may contain the variable `VAR` which obtains values from `RANGE`. For example, to specify connections to neighboring neurons, we can say

```
S.connect(j='i+(-1)**k for k in range(2)',
          skip_if_invalid=True)
```

where `skip_if_invalid` tells Brian to ignore invalid values for $j$ such as −1.

The simulators **NEURON** and **Arbor** do not support high-level connectivity rules and are therefore not included here.

The population-level connectivity rules shared—under different names—between two or more of the above simulators are the following:

**One-to-one** connects each source to one corresponding target.

**All-to-all** connects each source to all targets.

**Explicit connections** establishes the connections given in an explicit list of source-target pairs.

**Pairwise Bernoulli** performs a Bernoulli trial for each possible source-target pair. With a certain probability $p$, the connection is included.

**Random, fixed total number** establishes exactly $N_{syn}$ connections between possible sources and targets.

**Random, fixed in-degree** connects exactly $K_{in}$ sources to each target (where the same source may be counted more than once).

**Random, fixed out-degree** connects each source to exactly $K_{out}$ targets (where the same target may be counted more than once).

Languages and simulators vary with regard to whether autapses or multapses are created by a connectivity rule and whether it is possible to choose if they are created or not. Table 1 details the extent to which the rules above are implemented in the languages and simulators NEST, PyNN, NETPyNE, and NineML. In addition, PyNN supports the following rules:

- Pairwise Bernoulli with probability given as a function of either source-target distance, vector, or indices.

- Small-world connectivity of the Watts-Strogatz type, with and without autapses; out-degree can be specified.

- Connectivity specified by a CSA connection set provided by a CSA library.

- Explicit Boolean connection matrix.

- Connect cells with the same connectivity as the given PyNN projection.

**Table 1. Connectivity rules present in a selection of languages and simulators.** X: The rule is supported, A: The rule is supported and it is possible to specify whether autapses are created or not, M: Ditto for multapses.

|  | NEST | PyNN | NetPyNN | NineML |
|---|---|---|---|---|
| One-to-one | A | A |  | X |
| All-to-all | A | A | X[1] | X |
| Explicit connections | X[3] | X | X | X |
| Pairwise Bernoulli | A | A | X[1] | X |
| Random, fixed total number | AM | AM |  |  |
| Random, fixed in-degree | AM | AM | X[2] | X |
| Random, fixed out-degree | AM | AM | X[2] | X |

[1]. Autapses unconditionally included.

[2]. Neither autapses nor multapses are included.

[3]. Supported by passing lists to `Connect` and choosing the `one_to_one` rule.

The pairwise Bernoulli and random, fixed in- and out-degree rules in NEST support connectivity creation based on the relative position of source and target neurons.

## Connectivity concepts

We here provide formal definitions of connectivity concepts for neuronal network models. These concepts encompass the basic connectivity rules illustrated in Fig 7 which are already commonly used by the computational neuroscience community (see Fig 6). Beyond that, we discuss concepts to reflect some of the richness of anatomical brain connectivity and complement in particular non-spatial connectivity rules with rules for spatially organized connectivity.

For each high-level connectivity rule, we give both an algorithmic construction rule and the resulting connectivity distribution. Modelers can use these definitions to succinctly specify connection rules in their studies. However, if details differ from our standard definitions, these details should still be specified. Furthermore, we suggest symbols that can be used to indicate the corresponding connectivity types in network diagrams and add the corresponding CSA expressions from [58].

In the specification of connectivity concepts we use the following notations and definitions. Let $\mathcal{S} = \{s_1, \ldots, s_{N_s}\}$ be the ordered set of sources of cardinality $N_s$ and $\mathcal{T} = \{t_1, \ldots, t_{N_t}\}$ the set of targets of cardinality $N_t$. Then the set of all possible directed edges between members of $\mathcal{S}$ and $\mathcal{T}$ is given by the Cartesian product $\mathcal{E}_{ST} = \mathcal{S} \times \mathcal{T}$ of cardinality $N_s \cdot N_t$.

If the source and target populations are identical ($\mathcal{S} = \mathcal{T}$) a source can be its own target. We call such a self-connection an *autapse* (cf. Fig 7). If autapses are not allowed, the target set for any node $i \in \mathcal{S}$ is $\mathcal{T} = \mathcal{S} \setminus i$, with cardinality $N_t = N_s - 1$. If there is more than one edge between a source and target (or from a node to itself), we call this a *multapse*.

The *degree distribution $P(k)$* is the distribution across nodes of the number of edges per node. In a directed network, the distribution of the number of edges going out of (into) a node is called the *out-degree* (*in-degree*) distribution. The distributions given below describe the effect of applying a connection rule once to a given $\mathcal{S} - \mathcal{T}$ pair.

**Deterministic connectivity rules.** Deterministic connectivity rules establish precisely defined sets of connections without any variability across network realizations.

### One-to-one
**Symbol**: $\delta$
**CSA**: $\delta$
**Definition**: Each node in $\mathcal{S}$ is uniquely connected to one node in $\mathcal{T}$.
$\mathcal{S}$ and $\mathcal{T}$ must have identical cardinality $N_s = N_t$, see Fig 7A. Both sources and targets can be permuted independently even if $\mathcal{S} = \mathcal{T}$. The in- and out-degree distributions are given by $P(K) = \delta_{K,1}$, with Kronecker delta $\delta_{i,j} = 1$ if $i = j$, and zero otherwise.

### All-to-all
**Symbol**: $\Omega$
**CSA**: $\Omega$
**Definition**: Each node in $\mathcal{S}$ is connected to all nodes in $\mathcal{T}$.
The resulting edge set is the full edge set $\mathcal{E}_{ST}$. The in- and out-degree distributions are $P_{\text{in}}(K) = \delta_{K,N_s}$ for $\mathcal{T}$, and $P_{\text{out}}(K) = \delta_{K,N_t}$ for $\mathcal{S}$, respectively. An example is shown in Fig 7B.

### Explicit connections
**Symbol**: $X$
**CSA**: Not applicable

**Definition**: Connections are established according to an explicit list of source-target pairs. Connectivity is defined by an explicit list of sources and targets, also known as *adjacency list*, as for instance derived from anatomical measurements. It is, hence, not the result of any specific algorithm. An alternative way of representing a fixed connectivity is by means of the *adjacency matrix A*, such that $A_{ij} = 1$ if $j$ is connected to $i$, and zero otherwise. We here adopt the common computational neuroscience practice to have the first index $i$ denote the target and the second index $j$ denote the source node.

**Probabilistic connectivity rules.** Probabilistic connectivity rules establish edges according to a probabilistic rule. Consequently, the exact connectivity varies with realizations. Still, such connectivity leads to specific expectation values of network characteristics, such as degree distributions or correlation structure.

*Pairwise Bernoulli*
 **Symbol**: $p$
 **CSA**: $\rho(p)$
 **Definition**: Each pair of nodes, with source in $\mathcal{S}$ and target in $\mathcal{T}$, is connected with probability $p$.
 In its standard form this rule cannot produce multapses since each possible edge is visited only once. If $\mathcal{S} = \mathcal{T}$, this concept is similar to Erdős-Rényi-graphs of the *constant probability p*-ensemble $G(N, p)$—also called *binomial ensemble* [73]; the only difference being that we here consider directed graphs, whereas the Erdős-Rényi model is undirected. The distribution of both in- and out-degrees is binomial,

$$P(K_{\text{in}} = K) = \mathcal{B}(K|N_s, p) := \binom{N_s}{K} p^K (1 - p)^{N_s - K} \tag{7}$$

and

$$P(K_{\text{out}} = K) = \mathcal{B}(K|N_t, p), \tag{8}$$

respectively. The expected total number of edges equals $\mathrm{E}[N_{\text{syn}}] = pN_tN_s$.

*Random, fixed total number without multapses*
 **Symbol**: $N_{\text{syn}}, \cancel{M}$
 **CSA**: $\rho_{\mathbf{N}}\left(N_{\text{syn}}\right)(\mathbb{N}_S \times \mathbb{N}_T)$
 **Definition**: $N_{\text{syn}} \in \{0, \ldots, N_s N_t\}$ edges are randomly drawn from the edge set $\mathcal{E}_{ST}$ without replacement. For $\mathcal{S} = \mathcal{T}$ this is a directed graph generalization of Erdős-Rényi graphs of the *constant number of edges $N_{syn}$*-ensemble $G(N, N_{\text{syn}})$ [74]. There are $\binom{N_s N_t}{N_{\text{syn}}}$ possible networks for any given number $N_{\text{syn}} \leq N_s N_t$, which all have the same probability. The resulting in- and out-degree distributions are multivariate hypergeometric distributions.

$$P(K_{\text{in},1} = K_1, \ldots, K_{\text{in},N_t} = K_{N_t})$$

$$= \begin{cases} \prod_{j=1}^{N_t} \binom{N_s}{K_j} \Big/ \binom{N_s N_t}{N_{\text{syn}}} & \text{if } \sum_{j=1}^{N_t} K_j = N_{\text{syn}} \\ \\ 0 & \text{otherwise} \end{cases}, \tag{9}$$

and analogously $P(K_{\mathrm{out},1} = K_1, \ldots, K_{\mathrm{out},N_s} = K_{N_s})$ with $K_{\mathrm{out}}$ instead of $K_{\mathrm{in}}$ and source and target indices switched.

The marginal distributions, i.e., the probability distribution for any specific node $j$ to have in-degree $K_j$, are hypergeometric distributions

$$P(K_{\mathrm{in},j} = K_j) = \binom{N_s}{K_j} \binom{N_s N_t - 1}{N_{\mathrm{syn}} - K_j} \bigg/ \binom{N_s N_t}{N_{\mathrm{syn}}}, \tag{10}$$

with sources and targets switched for $P(K_{\mathrm{out},j} = K_j)$.

### Random, fixed total number with multapses

**Symbol**: $N_{\mathrm{syn}}$, $M$

**CSA**: $\rho_{\mathbf{N}}\left(N_{\mathrm{syn}}\right)\mathbf{M}(\mathbb{N}_S \times \mathbb{N}_T)$

**Definition**: $N_{\mathrm{syn}} \in \{0, \ldots, N_s N_t\}$ edges are randomly drawn from the edge set $\mathcal{E}_{ST}$ with replacement.

If multapses are allowed, there are $\binom{N_s N_t + N_{\mathrm{syn}} - 1}{N_{\mathrm{syn}}}$ possible networks for any given number $N_{\mathrm{syn}} \leq N_s N_t$. Because exactly $N_{\mathrm{syn}}$ connections are distributed across $N_t$ targets with replacement, the joint in-degree distribution is multinomial,

$$P(K_{\mathrm{in},1} = K_1, \ldots, K_{\mathrm{in},N_t} = K_{N_t})$$

$$= \begin{cases} \dfrac{N_{\mathrm{syn}}!}{K_1! \ldots K_{N_t}!} \, p^{N_{\mathrm{syn}}} & \text{if } \sum_{j=1}^{N_t} K_j = N_{\mathrm{syn}} \\[2em] 0 & \text{otherwise} \end{cases} \tag{11}$$

with $p = 1/N_t$.

The out-degrees have an analogous multinomial distribution $P(K_{\mathrm{out},1} = K_1, \ldots, K_{\mathrm{out},N_s} = K_{N_s})$, with $p = 1/N_s$ and sources and targets switched. The marginal distributions are binomial distributions $P(K_{\mathrm{in},j} = K) = \mathcal{B}(K|N_{\mathrm{syn}}, 1/N_t)$ and $P(K_{\mathrm{out},j} = K) = \mathcal{B}(K|N_{\mathrm{syn}}, 1/N_s)$, respectively.

The $\mathbf{M}$-operator of CSA should not be confused with the "$M$" indicating that multapses are allowed in our symbolic notation.

### Random, fixed in-degree without multapses

**Symbol**: $K_{\mathrm{in}}$, $\cancel{M}$

**CSA**: $\rho_1(K)(\mathbb{N}_S \times \mathbb{N}_T)$

**Definition**: Each target node in $\mathcal{T}$ is connected to $K_{\mathrm{in}}$ nodes in $\mathcal{S}$ randomly chosen without replacement.

The in-degree distribution is by definition $P(K) = \delta_{K,K_{\mathrm{in}}}$. To obtain the out-degree distribution, observe that after one target node has drawn its $K_{\mathrm{out}}$ sources the joint probability

distribution of out-degrees $K_{\text{out},j}$ is multivariate-hypergeometric such that

$$P(K_{\text{out},1} = K_1, \ldots, K_{\text{out},N_s} = K_{N_s})$$

$$= \begin{cases} \prod_{j=1}^{N_s} \binom{1}{K_j} \Big/ \binom{N_s}{K_{\text{in}}} & \text{if } \sum_{j=1}^{N_s} K_j = K_{\text{in}} \\ \\ 0 & \text{otherwise} \end{cases} \tag{12}$$

where $\forall_j K_j \in \{0, 1\}$. The marginal distributions are hypergeometric distributions

$$P(K_{\text{out},j} = K) = \binom{1}{K} \binom{N_s - 1}{K_{\text{in}} - K} \Big/ \binom{N_s}{K_{\text{in}}} = \text{Ber}(K_{\text{in}}/N_s), \tag{13}$$

with $\text{Ber}(p)$ denoting the Bernoulli distribution with parameter $p$, because $K \in \{0, 1\}$. The full joint distribution is the sum of $N_t$ independent instances of Eq 12.

### Random, fixed out-degree without multapses

**Symbol**: $K_{\text{out}}, \cancel{M}$

**CSA**: $\rho_0(K)(\mathbb{N}_S \times \mathbb{N}_T)$

**Definition**: Each source node in $\mathcal{S}$ is connected to $K_{\text{out}}$ nodes in $\mathcal{T}$ randomly chosen without replacement.

The out-degree distribution is by definition $P(K) = \delta_{K,K_{\text{out}}}$, while the in-degree distribution is obtained by switching source and target indices, and replacing $K_{\text{out}}$ with $K_{\text{in}}$ in Eq 12.

### Random, fixed in-degree with multapses

**Symbol**: $K_{\text{in}}, M$

**CSA**: $\rho_1(K)\mathbf{M}(\mathbb{N}_S \times \mathbb{N}_T)$

**Definition**: Each target node in $\mathcal{T}$ is connected to $K_{\text{in}}$ nodes in $\mathcal{S}$ randomly chosen with replacement.

$N_s$ is the number of source nodes from which exactly $K_{\text{in}}$ connections are drawn with equal probability $p = 1/N_s$ for each of the $N_t$ target nodes $t_i \in \mathcal{T}$. The in-degree distribution is by definition $P(K) = \delta_{K,K_{\text{in}}}$. To obtain the out-degree distribution, we observe that because multapses are allowed, drawing $N_t$ times $K_{\text{in},i} = K_{\text{in}}$ from $\mathcal{S}$ is equivalent to drawing $N_t K_{\text{in}}$ times with replacement from $\mathcal{S}$. This procedure yields a multinomial distribution of the out-degrees $K_{\text{out},j}$ of source nodes $s_j \in \mathcal{S}$ [75], i.e.,

$$P(K_{\text{out},1} = K_1, \ldots, K_{\text{out},N_s} = K_{N_s})$$

$$= \begin{cases} \dfrac{(N_t K_{\text{in}})!}{K_1! \ldots K_{N_s}!} p^{N_t K_{\text{in}}} & \text{if } \sum_{j=1}^{N_s} K_j = N_t K_{\text{in}} \\ \\ 0 & \text{otherwise} \end{cases} \tag{14}$$

The marginal distributions are binomial distributions

$$P(K_{\text{out},j} = K) = \mathcal{B}(K|N_t K_{\text{in}}, 1/N_s). \tag{15}$$

### Random, fixed out-degree with multapses

**Symbol**: $K_{\text{out}}, M$

**CSA**: $\rho_0(K)\mathbf{M}(\mathbb{N}_S \times \mathbb{N}_T)$

**Definition**: Each source node in $\mathcal{S}$ is connected to $K_{\text{out}}$ nodes in $\mathcal{T}$ randomly chosen with replacement.

By definition, the out-degree distribution is a $P(K) = \delta_{K,K_{\text{out}}}$. The respective in-degree distribution and marginal distributions are obtained by switching source and target indices, and replacing $K_{\text{out}}$ with $K_{\text{in}}$ in Eqs 14 and 15 [75].

**Networks embedded in metric spaces.** The previous sections analyze the connectivity between sets of nodes without any notion of space. However, real-world networks are often specified with respect to notions of proximity according to some metric. Prominent examples are spatial distance and path length in terms of number of intermediate nodes. The exact embedding into the metric space, such as the distribution of nodes in space $\rho(\vec{x})$ or the boundary conditions, can have a strong impact on the resulting network structure. $\rho$ here denotes the density, not to be confused with the CSA operator.

Given a distance-dependent connectivity, degree distributions result from this distance dependence combined with the distribution of distances between pairs of nodes [76]. If nodes are placed on a grid or uniformly at random in space, different asymptotic approximations to the degree distributions can be made [77–79]. If the node distribution $\rho(\vec{x})$ is (statistically) homogeneous, and the connection probability $p(\vec{x})$ is isotropic, the average in- or out-degree for connections to or from any node $i$ at a given distance $r = ||\vec{x}_i - \vec{x}||$ from the node follows $\langle K(r) \rangle \sim \rho(r)p(r)$, which is usually easier to derive than the full joint degree distribution, and can be used to statistically test whether network realizations are correctly generated [75, 80].

Here we specify the properties of spatial networks, which are also relevant for networks with feature-specific connectivity (e.g., based on sensory response tuning). In order to fully specify networks embedded in a metric space and with distance-dependent connectivity, the following quantities need to be listed:

**Dimension**: Most often the space is one-dimensional (e.g., ring networks), two-dimensional (e.g., a layer of neurons), or three-dimensional (i.e., a volume of neurons).

**Layout**: The layout $\rho(\vec{x})$ specifies how nodes are arranged, for instance on a regular grid (e.g., orthogonal, isometric, or hexagonal) or uniformly at random.

**Metric**: The metric specifies the concept of distance. On an orthogonal grid the max-norm metric ($\ell_\infty$) on the grid index can be the metric of choice, while for a uniformly random distribution of nodes the Euclidean metric ($\ell_2$) is typically chosen.

**Boundary conditions**: If nodes are embedded into a space with boundaries, there tend to be inhomogeneities in the connectivity close to these boundaries. To avoid such potential inconsistencies, boundary conditions are often assumed to be periodic, i.e., opposite borderlines are folded back onto each other (e.g., a line into a ring, a layer into a torus, etc.).

**Distance dependence of the connectivity profile**: The connectivity profile $f(\vec{x}_i, \vec{x}_j)$, sometimes called spatial footprint, specifies which nodes $j$ are connected to a node $i$ as a function of their distance $r_{ij} = ||\vec{x}_i - \vec{x}_j||$. Profiles can be deterministic (e.g., a node connects to all other nodes within a certain distance $r_{\text{max}}$, specified via a boxcar profile $f(r) \sim \Theta[r_{\text{max}} - r]$) or probabilistic (a node connects to another node at a certain distance $r$ with probability $p(r) \in [0, 1]$, e.g., boxcar: $p(r) \sim c\,\Theta[r_{\text{max}} - r]$, linear: $p(r) \sim \max(c_1 - c_2\,r, 0)$, sigmoidal: $p(r) \sim 1/(1 + e^{(r-c_1)/c_2})$, exponential: $p(r) \sim c_1\,e^{-r/c_2}$, Gaussian: $p(r) \sim c\,e^{-r^2/2\sigma^2}$, or more complex, e.g., non-centered multivariate Gaussian with covariance matrix $\Sigma$:

$$p(\vec{r}_{ij}) \sim c\,e^{-(\vec{r}_{ij}-\vec{\mu})^T \Sigma^{-1}(\vec{r}_{ij}-\vec{\mu})/2},\ \vec{r}_{ij} = \vec{x}_i - \vec{x}_j,\ \text{etc.}).$$ These distance-dependent connectivity

profiles may be combined with rules for the establishment of multapses and higher-order moments. In the case of feature-specific connectivity as well as other generalized spaces and cases where a metric is difficult to define, it can be useful to generalize $f$ to be a direct function of the sets of sources and targets, like a CSA mask: $f = f(i, j)$ where $i \in \mathcal{S}, j \in \mathcal{T}$. The distance could be treated similarly: $r_{ij} = r(i, j)$ corresponding to a CSA value function.

Larger-scale and multiscale networks can have more complicated, heterogeneous structures, such as layers, columns, areas, or hierarchically organized modules. Distance dependencies may then have to be specified with respect to the different levels of organization, for example specific to their horizontal (laminar) and vertical (e.g., columnar) dimension (cf. "Introduction"). One example is networks modeling axonal patches, i.e., neurons that have axonal arborization in a certain local range, as well as further axonal sprouting in several distinct long-range patches [24, 27, 81–83].

We discuss an explicit example of how to describe such connectivity rules in Section "Examples".

## Proposal for a graphical notation for network models

Network illustrations are a direct expression of how researchers think about a model and they are therefore a common means of network description (Fig 3B). They convey an intuitive understanding of network structures, relationships, and other properties central to the dynamics [84], and may also reflect how a model is implemented. If similar diagram styles are used, diagrams facilitate the reading of an article and allow for comparability of models across publications. However, computational neuroscience publications exhibit a wide variety of network diagram styles. While individual research groups and some sub-communities use similar symbols across publications, a common standard for the whole field has not been established yet.

In contrast, the related field of systems biology has developed the broadly accepted Systems Biology Graphical Notation (SBGN, [85]; see also [86]) over more than two decades. SBGN has an online portal (https://sbgn.github.io), an exchange and data format (SBGN-ML), a software library, and various further tools and databases.

Building on current practice in the computational neuroscience community, we propose a graphical notation framework for network models in computational neuroscience by defining which network elements to encode and how. We restrict ourselves to the simplest, most commonly used elements and provide a path to flexibly extend and customize the diagrams depending on the model specifics to expose. The notation uses simple standardized graphical components and therefore does not depend on a specific tool.

In the notation, a network is depicted as a graph composed of nodes and edges and enhanced with annotations. The nodes correspond to neuronal units or devices, the edges to connections, and the annotations specify the connections in terms of connection rules, possible constraints, and parameterization. The term "devices" refers to instances which are considered external to the main neuronal network but interact with it: either providing a stimulus or recording neuronal activity. Note that the nodes and edges of the graphical notation can combine multiple nodes and edges of the neuronal network; for instance, a population of network nodes can be indicated with one graphical node. A projection, referring to the set of connections resulting from one connectivity rule applied to a given source and target population, can be indicated with a single edge in the graphical notation.

Here we define diagram nodes and edges as well as annotations for the most common network types and propose a set of graphical elements to use. Thus, in the following, "node" and "edge" refer to the graphical components. A summarizing overview is given in Fig 8 for

## Network node

| Node class | |
| --- | --- |
| Individual unit | □ |
| Population | ▢ |
| **Node type** | |
| Generic node | □ |
| Excitatory neural node | △ |
| Inhibitory neural node | ◯ |
| Stimulating device node | ⬡ |
| Recording device node | ▱ |

## Network edge

| Determinism | |
| --- | --- |
| Deterministic | ——— |
| Probabilistic | - - - - - |
| **Edge type** | |
| Generic edge | ⟶ |
| Excitatory edge | ▸ |
| Inhibitory edge | ● |
| **Directionality** | |
| Unidirectional | ⟶ |
| Bidirectional | ⟷ |

## Annotation

| Connectivity concept | |
| --- | --- |
| **Concept** | |
| One-to-one | $\delta$ |
| All-to-all | $\Omega$ |
| Explicit connections | $X$ |
| Pairwise Bernoulli | $p$ |
| Random, fixed total number | $N_{\mathrm{syn}}$ |
| Random, fixed in-degree | $K_{\mathrm{in}}$ |
| Random, fixed out-degree | $K_{\mathrm{out}}$ |
| **Constraint** | |
| Autapses allowed | $A$ |
| Multapses allowed | $M$ |
| Prohibited | $\not{A}, \not{M}$ |
| **Parameterization** | |
| Constant parameter | $\overline{w}$ |
| Distributed parameter | $w \sim \mathcal{D}$ |
| **Further specification** | |
| Functional dependence | $f(\cdot)$ |

## Example

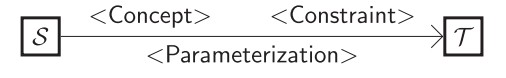

**Fig 8. Quick reference for the proposed graphical notation for network models in computational neuroscience.**

reference. The section concludes with a discussion on further techniques for creating appealing network diagrams.

**Network node.** A network node in the graphical notation represents one or multiple units. These units are either neuron or neural population models, or devices providing input or output. Network connectivity is defined between these graphically represented nodes. Nodes are drawn as basic shapes. A textual label can be placed inside the node for identification. Nodes are differentiated according to a node class and a node type.

**Node class.** The node class determines if a node represents an individual unit or a population of units by different frames of the shapes depicting the nodes. The distinction is a recommendation for diagrams that contain both kinds of nodes.

*Individual unit*

A node representing an individual unit may be depicted as a shape with a thin, single frame. Note that such an individual unit may be a population (e.g., neural mass) model.

*Population*

A node representing a population of units may be depicted as a shape with either a thick frame or a double frame. It is in principle possible to represent a group of population models this way.

**Node type.** The node type refers to a defining property of a node and is expressed by a unique shape.

*Generic node*

A generic node, represented by a square, is used if the specific node types do not apply or are not intended to be emphasized.

*Excitatory neural node*

An excitatory neural node, depicted by a triangle, is used if the units represent neurons, and their effect on targets is excitatory.

*Inhibitory neural node*

An inhibitory neural node, depicted by a circle, is used if the units represent neurons and their effect on targets is inhibitory.

*Stimulating device node*

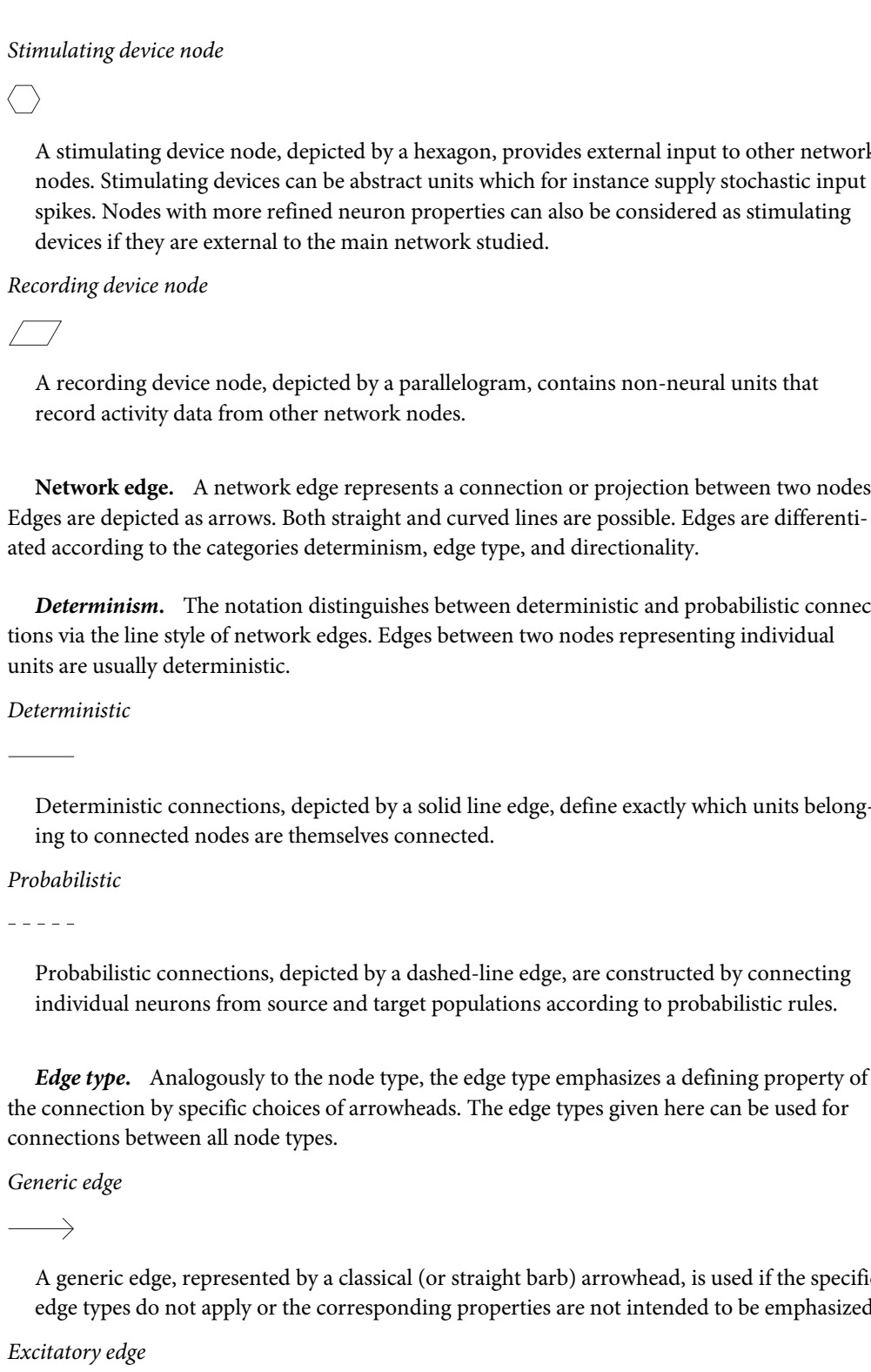

A stimulating device node, depicted by a hexagon, provides external input to other network nodes. Stimulating devices can be abstract units which for instance supply stochastic input spikes. Nodes with more refined neuron properties can also be considered as stimulating devices if they are external to the main network studied.

*Recording device node*

A recording device node, depicted by a parallelogram, contains non-neural units that record activity data from other network nodes.

**Network edge.**   A network edge represents a connection or projection between two nodes. Edges are depicted as arrows. Both straight and curved lines are possible. Edges are differentiated according to the categories determinism, edge type, and directionality.

**Determinism.**   The notation distinguishes between deterministic and probabilistic connections via the line style of network edges. Edges between two nodes representing individual units are usually deterministic.

*Deterministic*

Deterministic connections, depicted by a solid line edge, define exactly which units belonging to connected nodes are themselves connected.

*Probabilistic*

Probabilistic connections, depicted by a dashed-line edge, are constructed by connecting individual neurons from source and target populations according to probabilistic rules.

**Edge type.**   Analogously to the node type, the edge type emphasizes a defining property of the connection by specific choices of arrowheads. The edge types given here can be used for connections between all node types.

*Generic edge*

A generic edge, represented by a classical (or straight barb) arrowhead, is used if the specific edge types do not apply or the corresponding properties are not intended to be emphasized.

*Excitatory edge*

An excitatory edge, depicted by a triangle arrowhead, is used if the effect on targets is excitatory.

*Inhibitory edge*

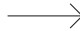

An inhibitory edge, depicted by a filled circle tip, is used if the effect on targets is inhibitory.

**Directionality.**   The directionality indicates the direction of signal flow by the location of one or two arrowheads on the edge.

*Unidirectional*

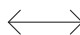

Unidirectional connections are depicted with a tip at the target node's end of the edge.

*Bidirectional*

$$\longleftrightarrow$$

Bidirectional connections are symmetric in terms of the existence of connections and their parameterization. Such connections are depicted with edges having tips on both ends. If the same units are connected but parameters for forward and backward connections are not identical, two separate unidirectional edges should be used instead.

**Annotation.**   Network edges can be annotated with information about the connection or projection they represent. Details on the rule specifying the existence of connections and their parameterization may be put along the arrow.

**Connectivity concept.**   The properties in this category further specify the presence or absence of connections between units within the connected nodes.

*Concept*
The definitions and symbols given in Section "Connectivity concepts" are the basis for this property.

*Constraint*
Specific constraint or exception to the connectivity concept.

  *Autapses allowed*
Autapses are self-connections. The letter $A$ indicates if they are allowed.

  *Multapses allowed*
Multapses are multiple connections between the same pair of units and in the same direction. The letter $M$ indicates if they are allowed.

  *Prohibited*
The symbol of a constraint struck out reverses allowed to prohibited. E.g., autapses and multapses are prohibited: $\not{A}, \not{M}$.

**Parameterization.**   Properties of the parameterization of connections, e.g., of weights $w$ and delays $d$, can be expressed with mathematical notation.

*Constant parameter*

A parameter, e.g., a weight, which takes on the same value for all individual connections is indicated by an overline: $\overline{w}$.

*Distributed parameter*

A tilde between a parameter (e.g., the weight) and a distribution indicates that individual parameter values are sampled from the distribution: $w \sim \mathcal{D}$. This example uses $\mathcal{D}$ for a generic distribution, but specific distributions, such as a normal distribution denoted by $\mathcal{N}$, are also possible.

***Further specification.*** Annotations for both the connectivity concept and the parameterization of connections can be specified further.

*Functional dependence*

Functional dependence on a parameter is expressed with parentheses, here indicated with a generic function $f(\cdot)$. Common use cases are the dependence on the inter-unit distance $r$ or on time $t$. Connections drawn with a distance-dependent profile can be indicated with $f(r)$. The exact function $f$ used should be defined close to the diagram; already defined concepts such as a spatially modulated pairwise Bernoulli connection probability can also be used: $p(r)$. Another example for a distance-dependent parameter could be a delay $d(r)$. Plastic networks, in which the weights change with time, can be indicated with $w(t)$.

**Customization and extension.** The definitions given above are intended as a reference for illustrating network types that are in the scope of this study. Further graphical techniques may be used that go beyond these fundamental definitions, such as adding meaning to the size of network nodes (e.g., making the area proportional to the population size) or using colors (e.g., to highlight network nodes or edges sharing certain specifics). In the community, two ways of distinguishing excitatory and inhibitory neurons tend to be used: the "water tap" notation in which the excitatory neurons are shown in red and the inhibitory neurons in blue (e.g., [87]), and notations in which the inhibitory neurons are shown in red and the excitatory neurons in either blue or black, which may be thought of as "bank account" notation (blue: [5], black: [14]).

Fig 7 uses the proposed symbols for generic node and edge types to demonstrate basic connectivity patterns; in addition, we employ colors to differentiate source and target nodes and their connections. In Fig 1 we distinguish with blue and red between excitatory and inhibitory neurons, respectively, to give an example for the bank account notation.

Encoding the same feature in multiple ways is also encouraged if it supports intuition; in the proposed graphical notation, we use double encoding for node shapes and arrowheads. For complex or hierarchical networks, multiple diagrams may be created: for instance, one that provides an overview and others that bring out specific details.

The modular structure of our graphical notation framework allows for extension to features that are not yet covered. Symbols for additional network elements may be defined for example in the figure legend and applied as the researcher sees fit. The common classification of neural nodes into excitatory and inhibitory types used in the notation is one such example. On the one hand, a model-specific definition of these types can be formulated. On the other hand, further classification detail can be added to the graph (e.g., in the form of annotations) or additional node types can be introduced if necessary to represent nodes with further biophysical properties which are not covered by the above simple classification.

In the same way as our propositions for node types can be customized, adjustment of the other graphical elements is also encouraged. For example, having so far considered only networks coupled via chemical synapses, another possible extension is to define gap junctions as a novel edge type. One possibility here is to use the common symbol for electrical resistance:

*Gap junctions*

∿∿∿

Electrical coupling via gap junctions is represented by a zig-zag line connecting the nodes.

## Examples

To illustrate the symbolic and graphical notation proposed, we apply it in the following to three concrete example networks.

**Two-population balanced random network.** The first example is the random, fixed in-degree variant of the balanced random network model also shown in Fig 1A (for details see Figs 12–15). Fig 9 shows different means for describing the connectivity of the model; the same options are covered in the model review in Fig 3B. The illustration (Fig 9A) uses the elements for nodes, edges, and annotations introduced in Section "Proposal for a graphical notation for network models" to depict the network composed of an excitatory (E, triangle) and an inhibitory (I, circle) neuron population, and a population of external stimulating devices (E$_{ext}$, hexagon). Recurrent connections between the neurons in the excitatory and inhibitory populations are probabilistic (dashed edges) and follow the "random, fixed in-degree" rule ($K_{in}$) with the further constraints that autapses are prohibited ($\cancel{A}$) and multapses are allowed ($M$).

Connections between different, non-intersecting populations by definition cannot have autapses and therefore it is not required to specify this along the corresponding edges. Neither does the absence of multapses between E$_{ext}$ and the neuronal populations need to be specified as we here assume a one-to-one connectivity ($\delta$). This network diagram not only indicates if connections exist but also shows that their parameters, weights ($w$), and delays ($d$) are the same for each connection. However, the diagram does not express the parameter values, just as the numbers of incoming connections are left to be defined elsewhere. In contrast, the textual description (Fig 9B) adds subscripts to the connectivity concept to indicate that the excitatory and inhibitory in-degrees may be different: $K_{in,\mathcal{T}E}$ and $K_{in,\mathcal{T}I}$, respectively. The table (Fig 9C) follows the guidelines by Nordlie et al. [84] and structures each connection in terms of a name, the source and target populations, and the connectivity rule. The set of equations (Fig 9D) formulates the connectivity by means of the Connection Set Algebra (CSA) [58]. While panels A–D of Fig 9 are primarily concerned with the conceptual description of connectivity, Fig 9E gives an implementation example using the PyNEST [63] interface of the simulator NEST [62]. The excitatory (E) and inhibitory (I) population are here represented by `NodeCollections`, storing the IDs of each neuron. By default, autapses and multapses are allowed; here we set both values explicitly for clarity. E$_{Ext}$ in the code stands for a `poisson_generator`, a stimulating device node in NEST which generates independent sequences of input spikes sampled from the same Poisson process for each of its target neurons. In other words, E$_{Ext}$ refers to just one NEST node which acts like the population of external stimulating devices indicated with E$_{Ext}$ in Fig 9A–9D. Due to this specific implementation of the `poisson_generator`, the default connection rule `all_to_all` as a generalization of one-to-all connectivity is here applied instead of `one_to_one`.

Previous studies preferentially combine different ways of describing connectivity (Fig 3B) and also the example in Fig 9 highlights that one means alone may not be sufficient to

## A Illustration

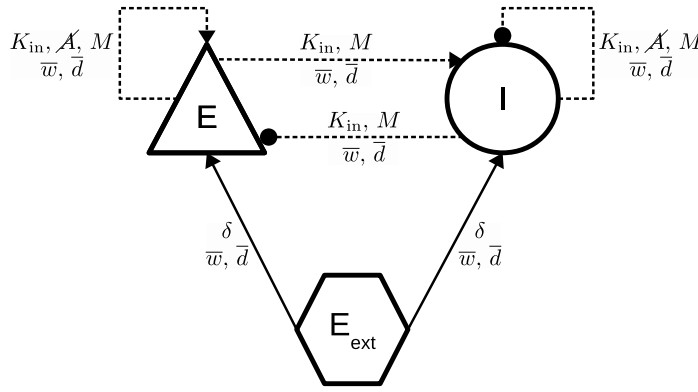

## B Text

Each neuron receives $K_{\text{in},\mathcal{T}\,\text{E}}$ excitatory connections randomly drawn from population E and $K_{\text{in},\mathcal{T}\,\text{I}}$ inhibitory connections from population I. Autapses are prohibited and multapses are allowed. Each neuron receives additional input from an external stimulating device.

## C Table

| Connectivity | | | |
|---|---|---|---|
| **Name** | **Source** | **Target** | **Pattern** |
| EE | E | E | Random, fixed in-degree $K_{\text{in},\mathcal{T}\,\text{E}}$ with multapses (autapses prohibited) |
| IE | E | I | Random, fixed in-degree $K_{\text{in},\mathcal{T}\,\text{E}}$ with multapses |
| EI | I | E | Random, fixed in-degree $K_{\text{in},\mathcal{T}\,\text{I}}$ with multapses |
| II | I | I | Random, fixed in-degree $K_{\text{in},\mathcal{T}\,\text{I}}$ with multapses (autapses prohibited) |
| Ext | $E_{\text{ext}}$ | E $\cup$ I | One-to-one |

## D Equation

$$C_{\mathcal{T}\,\text{E}} = \langle \rho_{\mathbf{1}}(K_{\text{in},\mathcal{T}\,\text{E}})\mathbf{M}(\text{E} \times (\text{E} \cup \text{I})) \rangle$$
$$C_{\mathcal{T}\,\text{I}} = \langle \rho_{\mathbf{1}}(K_{\text{in},\mathcal{T}\,\text{I}})\mathbf{M}(\text{I} \times (\text{E} \cup \text{I})) \rangle$$
$$C_{\text{Ext}} = \langle \text{E}_{\text{ext}} \times (\text{E} \cup \text{I}) \rangle$$

## E Code

```
nest.Connect(pre=E, post=E+I,
             conn_spec={'rule': 'fixed_indegree', 'indegree': K_in_TE,
                        'autapses_allowed': False, 'multapses_allowed': True})

nest.Connect(pre=I, post=E+I,
             conn_spec={'rule': 'fixed_indegree', 'indegree': K_in_TI,
                        'autapses_allowed': False, 'multapses_allowed': True})

nest.Connect(pre=E_ext, post=E+I,
             conn_spec={'rule': 'all_to_all'})
```

**Fig 9. Different means to describe connectivity of a balanced random network.** Example descriptions for the model used in Fig 1A with description means similar to Fig 3B. (A) Network diagram according to the graphical notation introduced in Section "Proposal for a graphical notation for network models". Symbols in annotations refer to the concepts and not the explicit parameters. (B) Textual description of the model layout. Subscript "$\mathcal{T}$ E" labels connections from source population E to target population $\mathcal{T} \in \{\text{E, I}\}$; the same applies to "$\mathcal{T}$ I" with source population I. $K_{\text{in},\mathcal{T}\,\text{E}}$ and $K_{\text{in},\mathcal{T}\,\text{I}}$ represent the explicit values used for the in-degrees. (C) Table according to the guidelines by Nordlie et al. [84]. (D) Equations according to the Connection Set Algebra (CSA) [58] using the index sets *E* and *I*. (E) PyNEST source code [63] specifying connections from source (`pre`) to target (`post`) populations with a connection dictionary (`conn_spec`). The use of all-to-all instead of one-to-one connectivity here is due to the specific implementation of the external drive in NEST.

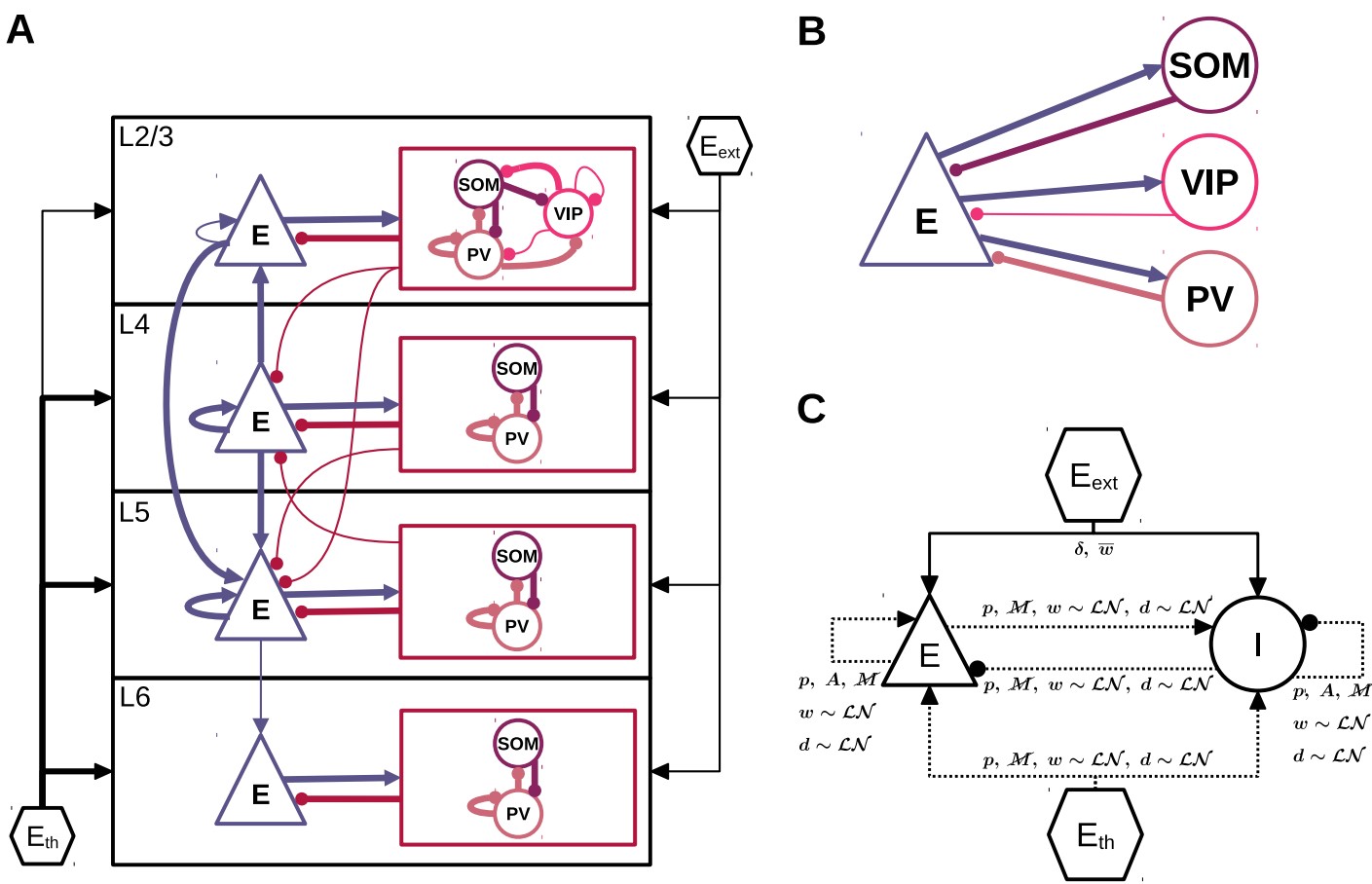

**Fig 10. Multi-layer microcircuit model with three inhibitory neuron types.** (A) Schematic overview of all neuronal populations, external inputs, and main connections. Inhibitory populations are grouped by boxes. In panels A and B, for probabilistic connections, only those with a probability of at least 4% are shown (thin lines: 4 to 8%, thick lines: ≥8%). (B) Detailed L2/3 connectivity between excitatory population and all three inhibitory populations; in panel A these connections are combined in two arrows (from and to the box). (C) Excitatory-inhibitory subnetwork with external inputs depicted with annotations according to the graphical notation in Fig 8. The connectivity is described with the rules "one-to-one" ($\delta$) and "pairwise Bernoulli" ($p$), and the constraints autapses allowed ($A$) and multapses prohibited ($\not{M}$). The synaptic weights ($w$) and delays ($d$) are specified as either constant (i.e., $\overline{w}$) or sampled from lognormal distributions (i.e., $w \sim \mathcal{LN}$). Interneuron types: somatostatin expressing (SOM), vasoactive intestinal peptide expressing (VIP), parvalbumin expressing (PV).

exhaustively cover all aspects of the connectivity. For a comprehensive description, we recommend using at least one network diagram and a textual description for rapidly conveying the network structure, and a table for providing details. In addition, default assumptions, e.g., the presence or absence of multapses, should be made explicit; this can be done in the text.

**Cortical microcircuit with distinct interneuron types.** The second example, shown in Fig 10, is a cortical microcircuit model [88] adapted from Potjans and Diesmann [89]. Extending the two-population network in Fig 9, this model comprises four cortical layers (L2/3, L4, L5, and L6). With its cell-type and layer-specific connectivity, the Potjans-Diesmann model represents the structure and dynamics of local cortical circuitry which is similar across different areas and species. The model has been used in a number of recent validation and benchmarking studies, and implementations for different simulators exist, including NEST [90], SpiNNaker [90, 91], Brian [92], GeNN and PyGeNN [93, 94], NeuronGPU [95], NetPyNE [96], and PyNN (available as a PyNN example and via Open Source Brain, https://www.

opensourcebrain.org/projects/potjansdiesmann2014). While the original model by Potjans and Diesmann has only one excitatory (E) and one inhibitory (I) neuron population per layer, the model considered here distinguishes between three different inhibitory neuron types (SOM, VIP, and PV). All neuron populations receive external Poisson input $E_{ext}$ as in Fig 9 and additional input from an external thalamic population $E_{th}$. The thalamic targeting of all layers is in contrast to the Potjans-Diesmann model where only L4 and L6 receive thalamic input. Fig 10 shows three different diagrams to emphasize different aspects of the model. Here, the first two panels are used to give an intuitive overview of the network, while the third panel adheres to the proposed graphical notation to unequivocally represent the connectivity rules. Fig 10A uses a colored illustration to convey the overall components without specifying the connection rules. For the general model overview, cortical layers and subnetworks of inhibitory populations are framed by boxes. To avoid clutter, not all connections are shown and the distinction between probabilistic and deterministic connections via dashed and solid lines, as suggested in Fig 8, is not applied. Instead, only connections above a threshold connection probability are shown with solid lines, and two levels of line thickness help to distinguish between low- and high-probability connections. By taking this freedom we illustrate that customizations remain possible for overview figures, as long as the network is unequivocally described in the remainder. Arrows to or from a box represent the average connection probabilities to or from the network nodes contained in the box. The average connection probability equals the expected total number of connections divided by the maximum number of possible connections while considering all involved pairs of populations. For example, the average connection probability from an excitatory population E to the inhibitory populations $\mathcal{I} = \{PV, SOM, VIP\}$ is given by:

$$p_{\mathcal{I}E} = \frac{\sum_{I \in \mathcal{I}} N_E N_I p_{IE}}{\sum_{I \in \mathcal{I}} N_E N_I}. \tag{16}$$

Fig 10B zooms into layer L2/3 to highlight the connectivity between the excitatory population and the three inhibitory populations in this single layer, resolving the arrows in and out of the box. In panel A there is only one outgoing arrow from the inhibitory neuron box in L2/3 connecting to the excitatory population, but in panel B it becomes clear that the inhibitory subpopulations SOM and PV both have strong connections to E while VIP does not.

Fig 10C follows the proposed notations, as in Fig 9A, to illustrate the general components and connection rules that apply to the whole network regardless of layer and inhibitory cell type. While the original model by Potjans and Diesmann uses connectivity of the type "Random, fixed total number with multapses", this model uses "pairwise Bernoulli" connectivity as indicated by the symbol $p$.

Combining these illustrations helps to understand the structure and characteristics of this model more intuitively. However, we do encourage deviations from and extensions to the proposed notation if it helps to improve the clarity of the diagrams, but these changes should be explained with care.

**Spatial network with horizontally inhomogeneous structure.** The third example is a network embedded into two-dimensional space introduced in a paper by Voges & Perrinet [97] to model the dynamics of neocortical networks with realistic horizontal connectivity. The "PB model", as it is called by the authors, incorporates both local and non-local connections between cells as observed for instance in the laminar structure of the visual cortex of cats [97, 98]. Local connectivity (footprint ≲ 150–300 $\mu$m) is observed to be approximately isotropic,

with nearby cells being more likely to be connected than cells farther apart. On longer scales ($\gtrsim$ 1mm) so-called patches can be observed where the axons sprout and form several connections in a confined area (see Introduction).

As mentioned in Section "Connectivity concepts", in order to define spatially embedded networks, the dimensions of the space, layout of neurons, metric of distances, boundary conditions, and, for distance-dependent connectivity, the form of this distance dependence need to be specified (Fig 11A). Here, $N_E$ excitatory and $N_I = N - N_E$ inhibitory neurons are embedded into a two-dimensional Euclidean space of size $[0, L) \times [0, L)$ with periodic boundary conditions (Fig 11A). Excitatory neurons are placed randomly according to a uniform distribution, while inhibitory neurons are distributed on jittered grid positions with

## A

### Spatial network definition

**Dimension:** two-dimensional space $[0, L) \times [0, L)$

**Layout:** E-pop: uniform random cell positions
I-pop: jittered grid positions ($\Delta = L/\sqrt{N_I}, \delta\vec{x} \sim \mathcal{U}[0, J]^2$)

**Metric:** Euclidean

**Boundary conditions:** periodic in both dimensions (torus)

**Distance dependence:**

**local connectivity:** $K_{\text{out}}^{\text{loc}} \sim \mathcal{B}\left[K_{\text{out}}^{\text{loc}}\middle|K_{\text{out,max}}^{\text{loc}}, p_{\text{out}}^{\text{loc}}\right]$

$$p_{\text{loc}}\left(||\vec{x} - \vec{x}_i||\middle|K_{\text{out}}^{\text{loc}}\right) = p_0 e^{-||\vec{x} - \vec{x}_i||^2/2\sigma^2}$$

**non-local connectivity:**

# patches/group: $\text{Np} \sim \mathcal{U}_{\mathbb{N}}[\text{Np}_{\text{min}}, \text{Np}_{\text{max}}]$
# patches/neuron: $\text{Npn} \sim \mathcal{B}[\text{Npn}|n_{\text{Npn}}, p_{\text{Npn}}]\Theta[\text{Np} - \text{Npn}]$

distance group to patch center:
$$\text{dp} = ||\vec{x}_g - \vec{x}_{p,k}|| \sim \mathcal{N}(\mu_{\text{dp}}, \sigma_{\text{dp}}^2), \ k \in \{0, \ldots, \text{Np}\}$$

relative angle: $\phi \sim \mathcal{U}[0, 2\pi)$

probability within patch:

$$K_{\text{out}}^{\text{patch}} \sim \mathcal{B}\left[K_{\text{out}}^{\text{patch}}\middle|K_{\text{out,max}}^{\text{patch}}, p_{\text{out}}^{\text{patch}}\right]$$

$$p_{\text{patch}}\left(||\vec{x} - \vec{x}_{p,k}||\middle|K_{\text{out}}^{\text{patch}}\right) = p_0 \Theta(\text{rp} - ||\vec{x} - \vec{x}_{p,k}||)$$

**total connection density of neuron $i$:**

$$c_{\text{total}}(i) = c_{\text{loc}}(i) + \sum_{k=1}^{\text{Npn}} c_{p,k}(i)$$

## B

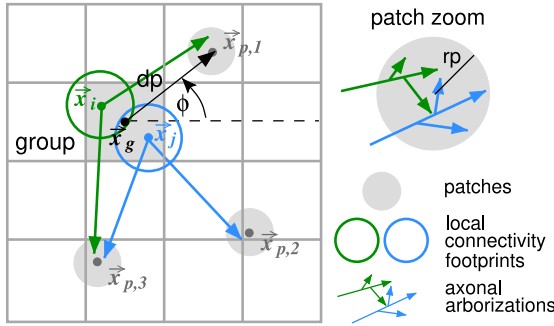

## C

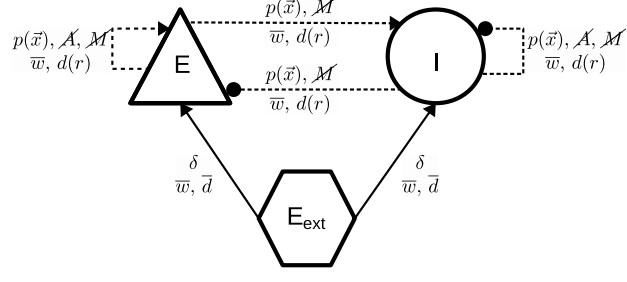

**Fig 11. Two-dimensional spatial network with patchy long-range connections.** (A) Spatial networks need to be defined in terms of dimension, layout, metric, boundary conditions, and the spatial or distance dependence of the connectivity, where applicable. In this example, neurons have both local and structured long-range connections [97]. $\Theta(x) = 0$ if $x < 0$, 1 otherwise. (B) Sketch of patchy connectivity and parameters needed to define $p_{\text{patch}}$. (C) Graphical notation of network connectivity corresponding to Fig 8.

grid constant $\Delta = L/\sqrt{N_I}$ and jitter $\delta\vec{x} \sim \mathcal{U}[[0, J] \times [0, J]]$ with maximal jitter $J$ where $\mathcal{U}$ denotes the uniform distribution. Both populations have local and non-local, patchy connections [97] with different parameterizations for excitatory and inhibitory neurons (Fig 11A), based on [23].

The global connection density (number of realized connections over number of possible connections) $c_{\text{total}}$ splits up into respective local and non-local parts motivated by anatomy [26, 97] (cf. Introduction), such that each neuron $i$ in a given subpopulation has local and non-local connection densities $c_{\text{loc}}(i)$ and $c_{\text{nonloc}}(i) = \sum_{k=1}^{\text{Npn}} c_{p,k}(i)$, where Npn is the number of patches per neuron (see Fig 11A). These underlying biologically motivated numbers then serve as constraints for the choices of the parameters needed in the following definitions [26, 97].

**Local connections**: In order to satisfy constraints with respect to both the fraction of connections assigned as local and the local spatial footprint, the out-degrees $K_{\text{out}}^{\text{loc}}$ are in a first step drawn from a binomial distribution with a mean that produces the right connectivity fraction $c_{\text{loc}}$. In a second step, random elements $(i, j)$ of the set of potential synapses are drawn, and a connection is established with probability $p_{\text{loc}}(||\vec{x}_j - \vec{x}_i|| | K_{\text{out},i}^{\text{loc}})$, until the required number $K_{\text{out},i}^{\text{loc}}$ is achieved. Multapses and autapses are excluded. The local connectivity of each neuron $i$ at $\vec{x}_i$ follows a Gaussian connectivity profile $p_{\text{loc}}(||\vec{x} - \vec{x}_i||)$ with center $\vec{x}_i$, maximal connection probability $p_0$ and space constant or footprint $\sigma$, indicated by colored circles in Fig 11B.

**Non-local, patchy connections**: Non-local connection patterns in Fig 11B are determined for groups of neighboring neurons, such that all neurons located within a certain region (squares) project to a fixed subset of spatially distributed patches (light gray disks) allowed for this region. Again, first an out-degree $K_{\text{out}}^{\text{patch}}$ is determined, and then the required number of synapses is established probabilistically according to Bernoulli trials, where the connection probability $p_{\text{patch}}(||\vec{x}_i - \vec{x}_{p,k}|| | K_{\text{out}}^{\text{patch}})$ from a neuron at $\vec{x}_i$ to each cell within one of its target patches centered at $\vec{x}_{p,k}$ is constant within a certain radius. Multapses are excluded.

The basic parameters to characterize patchy projections are then:

Np: the number of patches per group of neurons,

Npn: the number of patches per single neuron (Npn≤Np),

rp: the radius of a patch,

dp: the distance between the center of a group $\vec{x}_g$, and patch center $\vec{x}_{p,k}$,

$\phi$: the angle which characterizes a patch position $\vec{x}_{p,k}$ relative to $\vec{x}_g$, see Fig 11A and 11B.

In particular, here the respective Np's are drawn from uniform distributions with distinct minimum $\text{Np}_{\text{min}}$ and maximum $\text{Np}_{\text{max}}$ values for each population (E, I) while the Npn's (Npn≤Np) are drawn from binomial distributions $\mathcal{B}(\text{Npn}|n_{\text{Npn}}, p_{\text{Npn}})$ with specified means $\overline{\text{Npn}} = n_{\text{Npn}} p_{\text{Npn}}$ and the corresponding cutoff values. The distances dp come from normal distributions, while the angles $\phi$ are sampled uniformly from the interval $[0, 2\pi)$ (Fig 11A and 11B, [97]). The Npn patches to which a given neuron projects are chosen uniformly at random from the Np patches for the group.

The remaining connectivity specifications are shown in the graphic in Fig 11C. Each population receives an external drive modeled as Poisson-process spike input. Moreover, delays are distance-dependent [97]. The exact connectivity parameters for each population, as well as

weights and delays would need to be specified for instance in a table, which is beyond the scope of this example.

## Discussion

With the aim of supporting reproducibility in neuronal network modeling, we consider high-level connectivity in such models: connectivity that is described by rules applied to populations of nodes. As our main result, we propose a standardized nomenclature for often-used connectivity concepts and a graphical and symbolic notation for network diagrams.

Our proposal is informed by a review of model studies published in two well-known repositories (Open Source Brain and ModelDB), as representative of the wider body of neuronal network models. The network models reviewed are diverse in terms of when, where, and by whom they were published, their level of biological detail, and how their connectivity is defined and implemented (Figs 2–6). We find that the description of the connectivity in published articles is often insufficient for reproducing the connectivity rules, distributions, or concrete patterns used in the accompanying implementations. This is the case even though a large part of the identified connectivity concepts corresponds to rather basic rules. The devil is in the detail: deviations from standard rules, further constraints, or just the lack of rigorous definitions lead to ambiguities. Details sometimes omitted include whether self-connections (autapses) or multiple connections from a given source to a given target (multapses) are allowed.

In our review we further survey the use of high-level connection concepts in model descriptions and implementations and observe that they provide multiple advantages. High-level concepts allow for more concise and informative network specifications than explicit specification of atomic connectivity, in the form of either tables (databases) or algorithms expressed in elementary operations. Furthermore, for most high-level concepts presently used by modelers, simulation software provides dedicated code to efficiently instantiate connections in parallel or generate informative visualizations [54]. A significant obstacle to the systematic use of high-level concepts at present is the lack of a standardized terminology in the field: The same term may describe slightly different connectivity concepts with different authors or simulation codes, especially with respect to underlying assumptions about constraints, such as the presence of autapses.

In contrast to other approaches, we do not propose a new formal language (e.g., NeuroML, NineML) or a software implementation (e.g., PyNN). Instead, we gather terminology already in use in the community, expose interrelationships, and provide precise definitions. The result is a recipe helping neuroscientists to present their modeling work such that it has a higher chance of being reproducible. Furthermore, the user-level documentation of simulation engines can make reference to the presented definitions of connectivity concepts and point out any differences compared to the implementation at hand. A continuing debate and refactoring of individual codes may ultimately lead to a maturation of the field and the convergence of simulation engines.

### Historical context

For more than a decade, computational neuroscientists have been aware of the need to gather the notions used to describe the structure of network models and to establish common practices for network definitions [86]. Ideas on systematizing connectivity concepts were discussed at the "NEST Topology Library & LFP Modeling Workshop" in 2008 at the Norwegian University of Life Sciences (NMBU). This resulted in two publications on tabular [84] and

graphical [54] network representations. The workshop "Creating, Documenting and Sharing Network Models" held at the University of Edinburgh in 2011 reviewed the situation at the time and resulted in a joint article by the participants [55] which set out the research program for the present work.

Other efforts in the community have focused on implementations in specific tools or sets of tools. Examples include the simulation package NetPyNE [99], the model description languages NeuroML [44] and NineML [57], the SONATA data format for describing large-scale network models [59], and the Open Source Brain repository for network models also used here [43]. To foster the adoption, interoperability, and standardization of description languages and pertaining tools, the *INCF Working Group on Standardized Representations of Network Structures* (https://www.incf.org/sig/incf-working-group-standardized-representations-network-structures) was established in 2018.

While earlier work focused on the formal description of single neurons, network structure gained increasing importance. This was partly driven by the increasing complexity of network models but also by the need to reproduce network models of others. The latter is highlighted by the research on neuromorphic computing systems. Verification of these systems requires that the same network model can be instantiated on a conventional computer and on the new system under investigation. Groundbreaking work was carried out by the European FACETS project (2005–2010) in conceiving the meta-simulator language PyNN as a common front end for software and hardware simulation engines [46]. In this way, once a network has been formulated in PyNN, it can be instantiated in a software simulation engine such as NEST or a neuromorphic hardware system such as SpiNNaker [100]. The availability of high-level connectivity concepts such as "all-to-all" in PyNN must guarantee that all back-end engines interpret this in the same way. Expanding the connectivity concept into elementary pairwise connect requests already on the level of the PyNN interpreter is not an option as this would deprive a simulation engine of any chance of efficient parallelization. Another framework independent of a particular simulation engine is the Connection Set Algebra (CSA), described in Section "Description languages and simulators" [58].

Although fairly complete simulation codes for biological neuronal networks predate these decisive years by at least another decade [101–103], a framework for expressing connectivity that is consistent across simulators and description languages has not been developed to the present day. The primary reason is not that fundamental concepts such as cortical layers, random networks, and spatially organized networks only emerged over time. All these have been well known already for many decades and have been used in network models. In 2008, Erik De Schutter [86] analyzed the situation by comparing the fields of computational neuroscience and systems biology. He placed the emergence of computational neuroscience as a field in the 'second half of the eighties' and the emergence of systems biology in the 'late nineties' of the last century. Within a few years systems biology came up with a first version of the community standard SBML [104] for model description while computational neuroscience, although ten years older, was still struggling to find a common ground at the time of De Schutter's review. He explains the observation by a difference in scientific culture. Systems biology started on the background of large international collaborations for jointly uncovering the genome like the Human Genome Project (1990–2003). Thus researchers were already aware of the needs for standards and the methods to achieve them. Computational neuroscience only began to gain experience in large-scale collaborations with initiatives like the foundation of the Allen Institute for Brain Science in the US in 2003 and the European FACETS project in 2005, the latter eventually leading into the European Human Brain Project. Therefore, when the need for standardized model descriptions became apparent around 2010, the community was still learning

how to do big science. This may explain why it has taken so long to explore and discuss standardized model descriptions.

In addition, De Schutter points out, for the young discipline of systems biology modern software development tools and the idea of open source were part of the culture from the beginning. Therefore the competence in research software engineering and the acceptance of software development as an integral part of scientific methodology may have been more widespread at this time. There is a long-standing awareness that software development in science, including computational neuroscience [105, 106], is subject to special conditions: most scientists are not trained programmers [107], and it is often difficult to receive proper credit for the time invested in developing software [108]. As a result such tasks regularly are assigned low priority and progress is slow. It is the responsibility of senior scientists and science politics to adapt performance indicators to modern science and improve the conditions for sustainable research software engineering.

Although the present work restricts itself to a compilation of the concepts for describing network structure, we have learned from the history of SBML [104] and related efforts in computational neuroscience such as NeuroML [44] that it is important to integrate the different views of the community in a series of workshops. Chances for acceptance are higher if a proposed framework results from bottom-up experience and a community approach. Thus our results can only constitute a first draft which now needs to be discussed, elaborated, and maintained.

In systems biology it was customary to illustrate biochemical interactions by graphs as a third pillar of communication next to plain English and systems of equations in order to support the explanation of complex networks. Only a few years after the initial definition of SBML the idea emerged to also standardize the components of such illustrations as SBGN, a Systems Biology Graphical Notation [85]. Also in computational neuroscience researchers regularly communicate the structure of a model by illustrations of different styles and level of detail. While in systems biology the graphical notation expresses functional relations and temporal sequences depending on the diagram type of the standard, in computational neuroscience at present the primary use is the abstract representation of the anatomical connections of the neuronal network. In designing our draft graphical notation for computational neuroscience we tried to respect the lessons learned while developing SBGN as reported in [85]. In particular neither position nor color carry inherent meaning and we started from notations already used in the literature. Le Novère et al. point at the relevance of software tools using the graphical notation for dissemination. In this spirit, the recent release of NEST Desktop [109] already adheres to our proposed graphical notation.

## Limitations

The actual richness of models goes beyond the scope of this work and is still growing as the recent progress in experimental neuroscience makes more comprehensive anatomical and physiological data available to modelers. This data availability fuels the research field of connectomics and leads to an advent of large models with detailed data-driven connectivity [2, 3]. These models may have specific information not only on which neurons are connected but also on the location and other properties of the individual synapses. The models typically combine a bottom-up approach with conceptual assumptions. Abstractions are crucial for generalization and for testing hypotheses on the specifics of the connectivity. While the complexity of such models cannot be fully reduced, they may still benefit from guidelines for concise and reproducible descriptions of their connectivity.

Apart from complex, data-driven models, various high-level connectivity patterns exist which we have not discussed here. The connectivity rules used by modelers so far and considered here mostly yield regular and random graphs. In regular graphs, every node is linked to a fixed number of other nodes according to a standard pattern. In contrast, in random graphs all connections are established probabilistically. We have thereby neglected more complex topologies such as small-world networks, which are in between regular and random and are characterized by small short path lengths and a large clustering coefficient [110, 111]. Another example is scale-free networks, which are characterized by their power-law degree distribution. A small number of nodes (so-called "hubs") have a very high degree while most of the remaining ones have only few connections [73, 112]. Just as for data-driven models, future work may consider standards for consistently describing such networks.

Furthermore, the brains of many species, including mammals, follow a hierarchical organization, having different properties at different spatial scales. For instance, cerebral cortical areas are composed of layers, which contain populations of excitatory and inhibitory neurons, which may in turn be divided into subpopulations (cf. Fig 10). Many brain networks also have a clustered or modular structure, for example cortical networks consisting of macro- and mini-columns [113, 114]. This hierarchically modular organization suggests a multi-level description, where on the higher level not all details of the lower level are expressed for clarity. Our graphical notation already allows for nested populations, but the consistent description of hierarchical and modular networks requires further work.

Another aspect of biological neural networks we have neglected in this study is their adaptation over time via developmental processes and plasticity. Such plastic networks can for instance enable modeling inter-individual differences, potentially adding a layer of stochasticity beyond that of the initial structure. While the resulting networks are generally not easily captured in simple rules, compact accounts may be achieved by describing the initial state along with the growth or plasticity rules.

Neuronal network simulators should provide efficient high-level connectivity routines relevant for computational neuroscience. Which routines are available may, however, not solely depend on the need of the neuroscientist for a specific connection rule but also on the algorithmic efficiency. The rule "random, fixed total number", for instance, is non-trivial to parallelize [93]. Vice versa, which rules are already implemented in simulators may influence which ones neuroscientists eventually use. This relates to the general question how instruments shape the development of scientific theories [115–117]. Our literature review on published models shows that explicitly coded connectivity in general-purpose languages instead of using simulators with high-level commands is still quite common (Fig 4). A possible reason for this observation is that the effort to learn a simulator language outweighs a custom implementation as long as networks are small and the connectivity simple. The models in our review predominantly do not require a significant amount of computational resources and the chosen connectivity rules are not complicated to implement from scratch. We predict that the use of generic simulation codes will increase as models become more complex and the requirements for reproducible science and the publication of code become more strict. In turn this hopefully triggers an expansion of the simulators' repertoire of well-described and efficiently implemented connection routines. A challenge in this context is posed by the increasing use of high-performance computing facilities and specialized neuromorphic hardware [118, 119]. On future exascale supercomputers, highly efficient solutions for the parallel implementation of connectivity will be particularly important. Neuromorphic hardware is often constrained with regard to the neural network connectivity it supports, and the identification of relevant connectivity concepts can help decide which types of connectivity to enable. The concepts already

in use form a starting point for thinking about which high-level connectivity patterns future versions of simulation engines should provide.

## Perspectives

This work constitutes rather a starting point than an end point. Just as most existing network models, the concepts we describe are still limited with regard to connectivity structures observed in neuroanatomy. As models advance in capturing the complex multi-scale organization of the brain, this needs to be reflected in concepts and graphical notation such that researchers can always communicate on the appropriate level of resolution while having access to all details if needed. It is our hope that the methods laid down here help to structure the debate.

## Materials and methods

### Reviewed network models

Table 2 lists all articles included in the literature review in Section "Networks used in the computational neuroscience community".

### Balanced random network

The balanced random network model used in Figs 1 and 9 is based on the model introduced by Brunel [1]. Our implementation extends the script `brunel_delta_nest.py` which is part of the NEST source code (https://github.com/nest/nest-simulator) by the option to switch between a "fixed in-degree" and a "fixed out-degree" version. Details about the model description are summarized in Figs 12–14 and the parameters are given in Fig 15.

**Table 2. Alphabetical list of articles describing the reviewed network models.**

| | | | |
|---|---|---|---|
| Bartos et al. (2002) | [120] | Naze et al. (2015) | [121] |
| Brunel (2000) | [1] | Nicola and Clopath (2017) | [122] |
| Chauhan et al. (2018) | [123] | Pilly and Grossberg (2013) | [124] |
| Cohen (2014) | [125] | Potjans and Diesmann (2014) | [89] |
| Cutsuridis (2007) | [126] | Ramirez-Mahaluf et al. (2017) | [127] |
| del Molino et al. (2017) | [128] | Raudies et al. (2014) | [129] |
| Destexhe (2009) | [130] | Rennó-Costa and Tort (2017) | [131] |
| Gunn et al. (2017) | [132] | Sadeh et al. (2017) | [133] |
| Hu and Niebur (2017) | [134] | Stevens et al. (2013) | [135] |
| Huang et al. (2009) | [136] | Stroud et al. (2018) | [137] |
| Humphries and Gurney (2002) | [138] | Strüber et al. (2017) | [139] |
| Kazanovich and Borisyuk (2006) | [140] | Tikidji-Hamburyan and Canavier (2020) | [141] |
| Kuchibhotla et al. (2016) | [142] | Topalidou and Rougier (2015) | [143] |
| Kulvicius et al. (2008) | [144] | Ursino and Baston (2018) | [145] |
| Leblois (2006) | [146] | Vertechi et al. (2014) | [147] |
| Lian et al. (2019) | [148] | Vogels et al. (2011) | [149] |
| Machens et al. (2005) | [150] | Wang and Buzsáki (1996) | [151] |
| Masquelier and Kheradpisheh (2018) | [152] | Weber et al. (2006) | [153] |
| Masse et al. (2018) | [154] | Wystrach et al. (2016) | [155] |
| Mejias et al. (2016) | [156] | Yamazaki et al. (2015) | [157] |
| Morén et al. (2013) | [158] | Yang et al. (2016) | [159] |

| Model summary | |
|---|---|
| **Populations** | Three: excitatory, inhibitory, external input |
| **Connectivity** | <ul><li>"Fixed in-degree" version: random, fixed in-degree with multapses (autapses prohibited)</li><li>"Fixed out-degree" version: random, fixed out-degree with multapses (autapses prohibited)</li></ul> |
| **Neuron model** | Leaky integrate-and-fire (LIF), fixed voltage threshold, absolute refractory time (voltage clamp) |
| **Synapse model** | Static weights and delays, $\delta$-shaped current inputs (discontinuous voltage jumps) |
| **Input** | Independent fixed-rate Poisson spike trains to all neurons |
| **Measurement** | Spiking activity |

**Fig 12. Description of balanced random network models following the guidelines of Nordlie et al. [84].** Distinction between "fixed in-degree" and "fixed out-degree" versions.

| Populations | | |
|---|---|---|
| **Name** | **Elements** | **Size** |
| E | LIF neuron | $N_{\mathrm{E}} = 4N_{\mathrm{I}}$ |
| I | LIF neuron | $N_{\mathrm{I}}$ |
| $\mathrm{E_{ext}}$ | Poisson generator | $N_{\mathrm{E}} + N_{\mathrm{I}}$ |

| Connectivity | | | |
|---|---|---|---|
| **Name** | **Source** | **Target** | **Pattern** |
| $\mathcal{T}$E | E | $\mathcal{T} \in \{\mathrm{E,I}\}$ | <ul><li>"Fixed in-degree" version: random, fixed in-degree with multapses (autapses prohibited), $K_{\mathrm{in},\mathcal{T}\mathrm{E}} = \epsilon N_{\mathrm{E}}$</li><li>"Fixed out-degree" version: random, fixed out-degree with multapses (autapses prohibited), $K_{\mathrm{out},\mathcal{T}\mathrm{E}} = \epsilon(N_{\mathrm{E}} + N_{\mathrm{I}})$</li></ul> Connection strength $J$ |
| $\mathcal{T}$I | I | $\mathcal{T} \in \{\mathrm{E,I}\}$ | <ul><li>"Fixed in-degree" version: random, fixed in-degree with multapses (autapses prohibited), $K_{\mathrm{in},\mathcal{T}\mathrm{I}} = \epsilon N_{\mathrm{I}}$</li><li>"Fixed out-degree" version: random, fixed out-degree with multapses (autapses prohibited), $K_{\mathrm{out},\mathcal{T}\mathrm{I}} = \epsilon(N_{\mathrm{E}} + N_{\mathrm{I}})$</li></ul> Connection strength $-gJ$ |
| Ext | $\mathrm{E_{ext}}$ | $\mathrm{E} \cup \mathrm{I}$ | One-to-one, Connection strength $J$ |
| $\mathcal{T}\mathcal{S}$ | $\mathcal{S} \in \{\mathrm{E_{ext},E,I}\}$ | $\mathcal{T} \in \{\mathrm{E,I}\}$ | Conduction delay $d$ |

**Fig 13. Continuation of Fig 12.**

| Neuron and Synapse Model | |
|---|---|
| **Name** | LIF neuron |
| **Type** | Leaky integrate-and-fire, $\delta$-current input |
| **Subthreshold dynamics** | If $t > t^* + \tau_{\mathrm{ref}}$ $$\tau_{\mathrm{m}}\dot{V}(t) = -V(t) + RI(t)$$ $$I(t) = \frac{\tau_{\mathrm{m}}}{R}\sum_j J_j \delta(t - t_j^* - d)$$ with weight $J_j$, presynaptic spike time $t_j^*$ and conduction delay $d$ else $$V(t) = V_r$$ |
| **Spiking** | If $V(t-) < V_{\mathrm{th}} \wedge V(t+) \geq V_{\mathrm{th}}$ <br> 1. set $t^* = t$ <br> 2. emit spike with timestamp $t^*$ <br> 3. reset $V(t) = V_{\mathrm{r}}$ |

| Input | |
|---|---|
| Poisson generator | Independent fixed-rate Poisson spike trains to all neurons, $\nu_{\mathrm{ext}} = \eta V_{\mathrm{th}}/(\tau_{\mathrm{m}}J)$ |

| Measurements |
|---|
| Spiking activity as raster plots |

**Fig 14. Continuation of Fig 13.**

| Simulation | | |
|---|---|---|
| **Symbol** | **Value** | **Description** |
| $T_{\mathrm{sim}}$ | $1,000\,\mathrm{ms}$ | Simulation duration (first $200\,\mathrm{ms}$ not shown in raster plot) |
| $dt$ | $0.1\,\mathrm{ms}$ | Temporal resolution |
| $N_{\mathrm{rec}}$ | 50 | Number of neurons per population recorded from |

| Network | | |
|---|---|---|
| **Symbol** | **Value** | **Description** |
| $N_{\mathrm{I}}$ | 2,500 | Population size of inhibitory neurons |
| $d$ | $1.5\,\mathrm{ms}$ | Conduction delay |
| $J$ | $0.1\,\mathrm{mV}$ | Excitatory weight |
| $g$ | 5 | Ratio inhibitory weight / excitatory weight |
| $\epsilon$ | 0.1 | Connection probability |
| $\eta$ | 2 | External rate relative to threshold rate |

| Neuron model | | |
|---|---|---|
| **Symbol** | **Value** | **Description** |
| $E_{\mathrm{L}}$ | $0\,\mathrm{mV}$ | Resting potential |
| $V_{\mathrm{r}}$ | $0\,\mathrm{mV}$ | Reset potential |
| $V_{\mathrm{th}}$ | $20\,\mathrm{mV}$ | Firing threshold |
| $\tau_{\mathrm{m}}$ | $20\,\mathrm{ms}$ | Membrane time constant |
| $\tau_{\mathrm{ref}}$ | $2\,\mathrm{ms}$ | Absolute refractory period |
| $C_{\mathrm{m}}$ | $1\,\mathrm{pF}$ | Membrane capacitance |

**Fig 15. Simulation and network parameters.**

## Acknowledgments

The authors would like to thank Daniel Hjertholm for inspiring work on testing connectivity generation schemes, Sebastian Spreizer for immediately adopting the graphical notation in NEST Desktop, Espen Hagen for detailed comments on the manuscript, Hannah Bos for fruitful discussions, and our colleagues in the Simulation and Data Laboratory Neuroscience of the Jülich Supercomputing Centre for continuous collaboration. The authors gratefully acknowledge the computing time granted by the JARA Vergabegremium and provided on the JARA Partition part of the supercomputer JURECA at Forschungszentrum Jülich (computation grant JINB33).

## Author Contributions

**Conceptualization:** Johanna Senk, Birgit Kriener, Mikael Djurfeldt, Nicole Voges, Han-Jia Jiang, Lisa Schüttler, Gabriele Gramelsberger, Markus Diesmann, Hans E. Plesser, Sacha J. van Albada.

**Data curation:** Johanna Senk, Birgit Kriener, Mikael Djurfeldt, Nicole Voges, Han-Jia Jiang, Lisa Schüttler, Sacha J. van Albada.

**Formal analysis:** Johanna Senk, Birgit Kriener, Mikael Djurfeldt, Nicole Voges, Han-Jia Jiang, Lisa Schüttler, Sacha J. van Albada.

**Funding acquisition:** Johanna Senk, Gabriele Gramelsberger, Markus Diesmann, Hans E. Plesser, Sacha J. van Albada.

**Investigation:** Johanna Senk, Birgit Kriener, Mikael Djurfeldt, Nicole Voges, Han-Jia Jiang, Lisa Schüttler, Gabriele Gramelsberger, Markus Diesmann, Hans E. Plesser, Sacha J. van Albada.

**Methodology:** Johanna Senk, Birgit Kriener, Mikael Djurfeldt, Nicole Voges, Han-Jia Jiang, Lisa Schüttler, Gabriele Gramelsberger, Markus Diesmann, Hans E. Plesser, Sacha J. van Albada.

**Project administration:** Johanna Senk, Sacha J. van Albada.

**Resources:** Johanna Senk, Birgit Kriener, Mikael Djurfeldt, Nicole Voges, Han-Jia Jiang, Lisa Schüttler, Gabriele Gramelsberger, Markus Diesmann, Hans E. Plesser, Sacha J. van Albada.

**Software:** Johanna Senk, Birgit Kriener, Sacha J. van Albada.

**Supervision:** Gabriele Gramelsberger, Markus Diesmann, Hans E. Plesser, Sacha J. van Albada.

**Validation:** Johanna Senk.

**Visualization:** Johanna Senk, Birgit Kriener, Nicole Voges, Han-Jia Jiang.

**Writing – original draft:** Johanna Senk, Birgit Kriener, Mikael Djurfeldt, Nicole Voges, Han-Jia Jiang, Markus Diesmann, Hans E. Plesser, Sacha J. van Albada.

**Writing – review & editing:** Johanna Senk, Birgit Kriener, Mikael Djurfeldt, Nicole Voges, Han-Jia Jiang, Lisa Schüttler, Gabriele Gramelsberger, Markus Diesmann, Hans E. Plesser, Sacha J. van Albada.

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
