## [Decision Letter · Decision Letter 0]

14 Dec 2021

Dear Ms. Senk,

Thank you very much for submitting your manuscript "Connectivity Concepts in Neuronal Network Modeling" for consideration at PLOS Computational Biology.

As with all papers reviewed by the journal, your manuscript was reviewed by members of the editorial board and by several independent reviewers. In light of the reviews (below this email), we would like to invite the resubmission of a significantly-revised version that takes into account the reviewers' comments.

We cannot make any decision about publication until we have seen the revised manuscript and your response to the reviewers' comments. Your revised manuscript is also likely to be sent to reviewers for further evaluation.

Sincerely,

Roberto Toro

Associate Editor

PLOS Computational Biology

Daniele Marinazzo

Deputy Editor

PLOS Computational Biology

Dear Dr. Senk,

Thank you for submitting your manuscript. Two reviewers have now commented on your work, and they agree on its merits. I would be happy to accept your manuscript for publication after you take into account their remarks.

All the best,

Roberto Toro

Reviewer's Responses to Questions

**Comments to the Authors:**

Reviewer #1: This manuscript outlines a set of concepts, expressions and graphical entities which can be used to make network connectivity in models in computational neuroscience more explicit, reproducible and sharable. The ideas build on existing concepts used in simulators and published models and develops past work in this area by a number of the authors. The manuscript is well written and the concepts clearly presented, and can potentially form the basis of a consensus community approach to expressing and sharing such network models. The manuscript, while long, has no major flaws, but a few minor changes would improve it.

It would be good to have an explicit list (in an appendix perhaps) of the 42 network models considered in the "Networks used in the computational neuroscience community" section.

A bit more should be said about "devices" when they are introduced on page 22.

Re bold and plain outlines for Node classes - surely there is no need for making this distinction/requirement if there is only one class of node, i.e. either single units or just populations. The description here could be clarified.

There should be more discussion on what precisely defines an "excitatory" or "inhibitory" node. For simple neuronal models this is clear, but for more biophysically realistic circuits, glutamatergic or GABAergic synapses can act in the opposite way to the usual depending on the membrane potential of the post synaptic cell.

More should be said about electrically coupled networks containing gap junctions. These are often incorporated in networks even with single compartment cells, and it would just be a small amount of work to extend the discussion/graphical notation to incorporate these. They are often depicted as symmetric connections with flat/T shaped arrowheads on each end.

It should be clarified when Bidirectional connections are mentioned that in the case that there are 2 populations A&B with connections A->B and B->A, that 2 independent connections/arrows should be depicted (presumably with different parameters/conditions for each direction).

At line 961: autapses are prohibited and multapses allowed - therefore no M with line through it required here.

In Fig 8A, it is a bit difficult to see why Kin,EE, Kin,IE etc. aren't used in this depiction as opposed to just Kin 4 times, since 4 independent connections are shown. Also, the rest of the text implies Kin,EE and Kin,IE are independent values, whereas panel E suggests a single value K_in_TE is used for E->E and E->I.

Reviewer #2: Authors propose a methodology to fix a well known problem in computational neuroscience related to the (generally) ambiguous description of connectivity in neural models. Starting from an illustrative example, authors first review some terminology that is widely used in the computational neuroscience community, emphasizing and justifying the common usage of high level description for connectivity patterns. The introduction cites a lot of different studies to highlight the richness of connectivity patterns but I don't think this is really necessary (i.e. it is a well known fact). Authors should instead go straight to the question they're interested to answer without giving too broad considerations. For example, the paragraph on exascale is superfluous and I don't understand the link between a faithful description of connectivity and the ease of implementation at this stage. Furthermore, one important concept that may need to be introduced concerns the scalability of models: if your connectivity description is precise enough, you should be able to build a model with n=100 or n=1000 without much change in the results.

The introduction is followed by a small but detailed literature review (n=42) that exhibits the diversity of practices in both manuscripts and codes. While this literature is really interesting, I think it would deserve to be shorten since I think some data are not really relevant. For example, I don't think figure 2 brings anything useful for the debate and is also bit difficult to read, especially the Venn diagram that brings more confusion than it helps. These Venn diagrams are also used in other figures and bring equal confusion. For example, figure 3B is unreadable. Overall, I read this section as a detailed sequential description of the review results (which again are really good), but I would have expected a shorter and structured overview of practices and have put the actual details in the supplementary materials.

The section on languages and simulators is structured around procedural descriptions, declarative population-level descriptions, algebraic descriptions followed by a specific subsection about simulators and the different methods this used. This again a very detailed description (with even the name of methods for each simulator) that could be summarized withe a simple table and whether they implements this or that specific feature. As it is written, I find really hard to know which simulator does what.

The connectivity concepts section is clear and well structured, I do not have much comments but one on the subsection on network embedded in metric spaces. I wonder why authors do not generalize this specific case by considering the case where a function f: S (sources) x T (targets) -> R (reals) exists and could be used for defining the connectivity. If your network is embedded in a metric space, you have a "natural" function f with any distance function, but I can imagine cases where you do not have such metrics and yet there are specific relation between units (through the function f).

Based on these connection concepts, authors proposed a graphical notation to specify connectivity in a non ambiguous manner. While I appreciate the effort, I'm not totally convinced by the different examples that are given because I think there are really two different kinds of readers to consider. For example, figure 10 shows the patchy connection. On the one hand, figure 10b allows me to immediately get a sense of the connectivity pattern even though there is not enough information to implement it. On the other hand, figure 10c allows me to implement the patchy connection, but I do not get a sense of what it actually "looks like". In the end, I feel we may need both: approximate graphical representations for humans, and precise graphical representations for computers. Consequently, it would be worth to add an advice from the authors on when and how to use such graphical representations in an article. I've also one small question on the necessity of having a specific representation of inhibitory neurons knowing that you also have a specific representation for inhibitory edge. What would be the difference between an excitatory source node sending a inhibitory edge and an inhibitory source node sending an inhibitory edge?

Overall, I think this article is important and is worth to be published but I would advised authors to either drastically restructure/shorten (in order to focus on the main proposal) or to split the article in two articles (I & II in the same journal). The reason is that I think there are really two articles in one. One is about a review of practices in computational neurosciences based on a literature review and simulators implementation, with the various implications in term of reproducibility. This is already an interesting review and could be complemented by a state of the art in that domain. The second article is precisely a proposal for a generic and non-ambiguous connectivity description, supported by several examples and could be further extended with supplementary figures (for example, all the variation of an ambiguous textual description). Note that I'm not asking for such additions in the present article but only if there were two different articles.

Finally, I suggested some modifications in the manuscript in hope it would improve it but I'm perfectly fine if authors prefer not to perform such modifications as long as they explain why they prefer not to change.

**Have the authors made all data and (if applicable) computational code underlying the findings in their manuscript fully available?**

Reviewer #1: Yes

Reviewer #2: Yes

PLOS authors have the option to publish the peer review history of their article (what does this mean?). If published, this will include your full peer review and any attached files.

Reviewer #1: No

Reviewer #2: No
---

## [Decision Letter · Decision Letter 1]

7 Apr 2022

Dear Ms. Senk,

We are pleased to inform you that your manuscript 'Connectivity Concepts in Neuronal Network Modeling' has been provisionally accepted for publication in PLOS Computational Biology.

Best regards,

Roberto Toro

Associate Editor

PLOS Computational Biology

Daniele Marinazzo

Deputy Editor

PLOS Computational Biology

Reviewer's Responses to Questions

**Comments to the Authors:**

Reviewer #1: All my comments have been addressed in the updated manuscript, and I am happy for it to be published in its current form, apart from the minor point below.

NetPyNE is misspelled in Table 1.

Reviewer #2: I'm satisfied with authors answer and proposed modifications. One minor comment for the node class / node edge types on page 23 & 24. I would put symbols on the left next to the text and possibly using a light background colored box (with margin) such as to make symbols more visually salient. As they are currently, they are visible but less salient.

**Have the authors made all data and (if applicable) computational code underlying the findings in their manuscript fully available?**

Reviewer #1: Yes

Reviewer #2: Yes

PLOS authors have the option to publish the peer review history of their article (what does this mean?). If published, this will include your full peer review and any attached files.

Reviewer #1: No

Reviewer #2: No

---

## [Editor Report · Acceptance letter]

2 May 2022

PCOMPBIOL-D-21-01816R1 

Connectivity Concepts in Neuronal Network Modeling

Dear Dr Senk,

I am pleased to inform you that your manuscript has been formally accepted for publication in PLOS Computational Biology. Your manuscript is now with our production department and you will be notified of the publication date in due course.

With kind regards,

Livia Horvath
